
# Regional modelling of extreme sea levels induced by hurricanes

Alisée A. Chaigneau[1], Melisa Menéndez[1], Marta Ramírez-Pérez[1], Alexandra Toimil[1]

[1]IHCantabria - Instituto de Hidráulica Ambiental de la Universidad de Cantabria, Santander, Spain.

*Correspondence to:* Alisée A. Chaigneau (alisee.chaigneau@gmail.com)

5 **Abstract.** Coastal zones are increasingly threatened by extreme sea level events. Storm surges are one of the most hazardous components of these extremes, especially in regions prone to tropical cyclones. This study aims to explore factors affecting the performance of numerical modelling in simulating storm surges in the tropical Atlantic region. The maxima, duration and time evolution of the extreme storm surge events are evaluated for four historical hurricanes by comparison against tide gauge records. The ADCIRC and NEMO ocean models are intercompared using a similar configuration in terms of domain, bathymetry and spatial resolution. These models are then used to perform sensitivity experiments on oceanic and atmospheric forcings, physical parameterizations for wind stress and baroclinic/barotropic modes. NEMO and ADCIRC show a similar skill to simulate storm surges induced by hurricanes. Storm surges simulated with ERA5 atmospheric reanalysis forcing are generally more accurate than those using parametric wind models for simulated hurricanes. The inclusion of the baroclinic processes improves storm surge amplitudes in some coastal locations such as along the southeastern Florida peninsula (USA). Experiments exploring different wind stress implementations and the interactions between storm surges, tides and mean sea level however have shown a minimal impact on storm surges induced by hurricanes.

## 1. Introduction

Coastal zones are among the most densely populated and urbanized areas in the world. 10% of the world population lives in low-lying coastal regions with 35 million people in North America, Central America and the Caribbean region (McMichael et 20 al., 2020; Neumann et al., 2015). These regions are increasingly threatened by extreme sea levels, during which major damage to the waterfront and infrastructure is likely to occur (Hicke et al., 2022; Castellanos et al., 2022).

Tropical cyclones are major drivers of these extreme sea levels due to large storm surges, which are rises in the sea level due to the combined effect of low atmospheric pressure and strong winds (Woodworth et al., 2019). This phenomenon can drive coastal hazards such as flooding and erosion (Dullaart et al., 2021; Jamous et al., 2023). The present study focuses on four 25 historical severe tropical cyclones (hurricanes) that occurred in the northwestern Atlantic region in the last 20 years and have severely impacted the coasts: Wilma (2005), Matthew (2016), Irma (2017), and Maria (2017). Wilma is the most intense Atlantic hurricane by lowest pressure on record, formed on October 15, 2005, reaching sustained winds of 295 km/h before making landfall in southwestern Florida on October 24, 2005. Matthew, has formed on September 28, 2016, causing generalized devastation across the Caribbean and southeastern United States, particularly in Haiti, Cuba, and the Bahamas, 30 before weakening and dissipating over the Atlantic Ocean. Irma is also one of the strongest Atlantic hurricanes, formed on August 30, 2017, devastating several Caribbean islands and hitting Florida on September 10, 2017. Maria has formed on September 16, 2017, causing widespread destruction in Dominica and Puerto Rico before dissipating on September 30, 2017. As Category 5 hurricanes, all of them are major hurricanes for the region, both in terms of storm surge amplitudes reached and total damage estimated (Tab. 1).




| Hurricane | Time period considered | Number of hours in category 5 (i.e. > 70 m/s or 252 km/h) | Maximum storm surge level reported (m) | Affected zones | Total damage reported |
|---|---|---|---|---|---|
| Wilma | 2005/10/17 - 2005/11/08 | 3 | 3.7 | northeastern Yucatan Peninsula, western Cuba, southern Florida (USA), western Bahamas | 33 direct deaths $21 billion USD |
| Matthew | 2016/09/28 - 2016/10/11 | 2 | 3.9 | Haiti, southwestern Dominican Republic, Eastern Cuba, Bahamas, eastern Florida (USA) | 585 direct deaths 18 indirect deaths (USA), 128 persons missing (Haiti) $15 billion USD |
| Irma | 2017/08/30 - 2017/09/12 | 17 | 3.5 | All the northern Caribbean Islands, all Florida (USA) | 47 direct deaths 82 indirect deaths (USA) $53 billion USD |
| Maria | 2017/09/16 - 2017/09/29 | 7 | 2.9 | Puerto Rico, Virgen Islands, western Dominican Republic, Dominica, Guadeloupe | 3,000 direct and indirect deaths $92 billion USD |

**Table 1: Information about the selected hurricanes from the tropical cyclone reports of the National Hurricane Center (Pasch et al., 2006 for Wilma, Stewart, 2017 for Matthew, Cangialosi et al., 2021 for Irma and Pasch et al., 2023 for Maria).**

It is therefore important to monitor the spatio-temporal evolution of storm surges, especially in the context of climate change where tropical cyclone frequency and intensity might be altered (Roberts et al., 2020; Cattiaux et al., 2020; Knutson et al.,
2020; Bloemendaal et al., 2022; van Westen et al., 2023). For this purpose, historical records such as tide gauge data are valuable, but they are often scarce and sometimes unavailable during the most severe events (Haigh et al., 2021). Hydrodynamic models (e.g., ADCIRC, SCHISM, GTSM, Mike21) can be used to overcome this limitation. These models are often run in a 2-D barotropic mode enabling fine resolution along coastlines, thus limiting computational costs. In recent years, they have been widely used in operational systems to forecast storm surge hazards (Dietrich et al., 2018; Fernández-Montblanc
et al., 2019) or to generate regional (Haigh et al., 2014; Marsooli and Lin, 2018; Muis et al., 2019; Toomey et al., 2022; Gori et al., 2023; Martín et al., 2023; Parker et al., 2023) and global hindcasts (Muis et al., 2016; Dullaart et al., 2021). More recently, hydrodynamic models have also been employed to derive projections of storm surges at both regional (Camelo et al., 2020; Makris et al., 2023; Wood et al., 2023) and global (Vousdoukas et al., 2018; Muis et al., 2020, 2023) scales, driven by climate model data.

Primary drivers for hydrodynamic models are atmospheric forcings such as winds and atmospheric surface pressure. The use of a global or regional atmospheric reanalysis (e.g., ERA5, CFSR, JRA-55) provides a consistent hourly 2-D forcing field over the whole domain. However, in addition to unresolved processes and insufficient spatial resolution (Roberts et al., 2020), these datasets are limited in their temporal coverage, posing a challenge for hindcast production given the rarity of tropical cyclones (Dullaart et al., 2021; Wood et al., 2023). Parametric wind models, derived from observations or statistical approaches, enable
to compute a large number of simulations, thereby enhancing the robustness in storm surges evaluation (Haigh et al., 2014; Toomey et al., 2022; Martín et al., 2023). However, these parametric models often rely on simplifying cyclone behavior, frequently adopting an axisymmetric cyclone model, such as in the widely used Dynamic Holland Model (Holland, 1980; Fleming et al., 2008) potentially resulting in biases (Dietrich et al., 2018).

In addition to the atmospheric forcing, oceanic drivers are also important for storm surge modelling. The consideration of other
factors influencing sea level and their interactions, such as tides and regional mean sea level, can significantly modify storm surges (Marsooli and Lin, 2018; Idier et al., 2019). For instance, neglecting tide-surge interactions can significantly reduce the


accuracy in storm surge prediction (Fernández-Montblanc et al., 2019), potentially overestimating extreme sea levels by up to 30% (Arns et al., 2020). Other studies emphasize the importance of considering the baroclinic response in sea level due to tropical cyclones (Ezer, 2018; Zhai et al., 2019; Ye et al., 2020). 3-D baroclinic ocean general circulation models such as

NEMO and ROMS can be used for this purpose. These models explicitly resolve storm surges (Chaigneau et al., 2022; Irazoqui Apecechea et al., 2023) although at higher computational expenses. Their application in modelling storm surges due to tropical cyclones therefore remains limited (Kodaira et al., 2016; Hsu et al., 2023). Additionally, recent research highlights the significant impact of wind stress consideration on storm surge modelling, including the choice of the parameterization, the parameter tuning, and the impact of the processes considered such as waves (O'Neill et al., 2016; Pineau-Guillou et al., 2020).

This study aims to investigate different factors influencing the performance of numerical modelling in simulating storm surges caused by hurricanes. The focus is on the tropical Atlantic region, covering the Caribbean Sea, the Gulf of Mexico and eastern coasts of Florida (USA). Four historical hurricanes that have caused severe coastal impacts are simulated (Tab. 1). The skill of the simulations to reproduce the storm surge contribution to extreme sea levels is evaluated against recorded values from tide-gauge stations. The modelled peak surge maxima and the hourly time series are analyzed during these extreme events.

Two ocean models (ADCIRC and NEMO) are intercompared using a similar configuration: domain, spatial resolution of 9 km, bathymetry and 2-D barotropic mode. These models are then used to perform sensitivity experiments. The sensitivity of the atmospheric forcing is assessed by comparing storm surges induced by ERA5 reanalysis data and parametric wind models usually applied for hurricanes. The effect on storm surge due to non-linear interactions with the astronomical tide and variations in mean sea level is also investigated, as well as the sensitivity to different wind stress schemes. In addition, the baroclinic

contribution to storm surges is studied using a 3-D configuration that also simulates temperature and salinity and their impact on ocean circulation.

The remaining of the paper is organized as follows. The met-ocean data are presented in Sect. 2. The methods are described in Sect. 3, with details of the numerical models, configurations developed and sensitivity experiments performed, as well as the statistical metrics used to analyze the simulations. The results are presented in Sect. 4, first with a comparison of the models

with equivalent settings, then with an analysis of the sensitivity experiments. We notably examine the influence of atmospheric forcing using ADCIRC and the effect of oceanic drivers using NEMO. The results are discussed in Sect. 5 and general conclusions of the study are drawn in Sect. 6.

## 2. Met-ocean data

The modelled storm surges are validated against tide gauge records extracted from the GESLA (Global Extreme Sea Level

Analysis) dataset version3 (Haigh et al., 2023). The selected tide gauge stations provide high-frequency tide gauge records with at least an hourly frequency. In this study, tide gauges within a 300 km radius of the hurricanes are selected to analyze modelled storm surges. Their locations are listed in Table 2. Given the horizontal resolution of the regional models used, tide gauges located in onshore locations such as estuaries, channels, bays and lagoons are not considered in this study. The tidal harmonic constituents are extracted from the time series with the Python "utide" package (Codiga, 2011). The term "storm

surge" will be used hereinafter to denote the non-tidal residuals. Tide gauges registering storm surges of less than 15 cm are also excluded from the analysis.

| Tide gauge name | Country | Longitude (ºW) | Latitude (ºN) | Wilma | Matthew | Irma | Maria |
|---|---|---|---|---|---|---|---|
| Cedar_Key | USA | 83.0317 | 29.135 | | | x | |
| Crystal_Rv_At_Mouth_Nr_ Shell_Isl_Nr_Crystal_Rv_Fl | USA | 82.6906 | 28.9253 | | | x | |
| Gulf_Of_Mexico_Near_Bay port_Fl | USA | 82.6501 | 28.5336 | | x | x | |





| | | | | | | | |
|---|---|---|---|---|---|---|---|
| Clearwater_Bch_FL | USA | 82.832 | 27.977 | | | x | |
| Naples_FL | USA | 81.807 | 26.13 | x | x | x | |
| Key_West_FL | USA | 81.808 | 24.553 | x | | x | |
| Virginia_Key_FL | USA | 80.162 | 25.732 | x | | x | |
| Lake_Worth_Pier | USA | 80.0342 | 26.6128 | | x | x | |
| Trident_Pier | USA | 80.5931 | 28.4158 | x | x | x | |
| Punta_Cana | Dominican Republic | 68.375 | 18.505 | | | x | x |
| Mona_Island | Puerto Rico (USA) | 67.9385 | 18.0899 | | | x | x |
| Mayaguez_PR | Puerto Rico (USA) | 67.16 | 18.22 | | | x | |
| Yabucoa_Harbor_PR | Puerto Rico (USA) | 65.832 | 18.055 | | | | x |
| Fajardo_PR | Puerto Rico (USA) | 65.63 | 18.335 | | | x | |
| San_Juan_PR | Puerto Rico (USA) | 66.117 | 18.46 | | | x | x |
| Esperanza | Puerto Rico (USA) | 65.4714 | 18.0939 | | | x | |
| Isabel_Segunda | Puerto Rico (USA) | 65.4439 | 18.1525 | | | x | x |
| Lameshur_Bay_VI | Virgin Islands (USA) | 64.723 | 18.317 | | | x | |
| PointeAPitre_60minute | Guadeloupe (France) | 61.5300 | 16.23 | | | | x |

**Table 2: Selected tide gauge stations used for the storm surge validation.**

Storm surges induced by hurricanes are simulated using winds and pressure from the ERA5 reanalysis (Fig. 1b) provided by the European Centre for Medium-range Weather Forecasts (ECMWF) (Hersbach et al., 2020). The atmospheric variables have a horizontal resolution of 0.25 º and an hourly temporal resolution, covering the period from 1950 to the present. The assimilation of satellite data since 1980 enables the representation of tropical cyclones in the reanalyses. Compared to its predecessor ERA-Interim (79 km, 6-hourly), ERA5 higher spatial and temporal resolution allows for an improved resolution of tropical cyclones, for instance including lower central pressure (Hersbach et al., 2020). Additionally, ERA5 benefits from an improved data assimilation procedure, notably incorporating satellite observations from the Advanced Scatterometer (ASCAT) for wind speed (Dullaart et al., 2020). As a result, ERA5 has demonstrated improved representation of storm surges induced by tropical cyclones (Dullaart et al., 2020), leading to a recent extensive use in large-scale studies simulating storm surges (Muis et al., 2020, 2023; Dullaart et al., 2021; Gori et al., 2023; Parker et al., 2023).

Storm surges induced by hurricanes are also simulated using parametric wind models. These models represent the wind field distributions of tropical cyclones based on a limited number of observations, making them powerful tools due to their simplicity and computational efficiency. In our study, the tropical cyclone observations are taken from the International Best Track Archive for Climate Stewardship (IBTrACS) database (Knapp et al., 2010, 2018). It provides at least six-hourly information on the cyclone position and intensity from 1851 to present, although additional variables (such as radius of maximum wind, environmental pressure, and various wind radii) are also available for the last decades. The origin of parametric wind models started with CE. Deppermann (1947), which adopted the mathematical equations of the Rankine vortex model (Rankine, 1882) to depict the tropical cyclone atmospheric structure. Since then, numerous parametric models have been developed, becoming

more sophisticated and complex as increasing the observational technologies. This study evaluates four different parametric wind models. The first one, the Dynamic Holland Model (DHM), derives from the commonly used Holland profile (Holland, 1980), with the modifications applied by Fleming et al. (2008), to better capture dynamic processes within and around storms. Another evaluated Holland-derived model is that proposed by Willoughby et al. (2006). This is a more complex model based

on a piecewise continuous wind profile developed using an extensive aircraft data for validation purposes. On the other hand, we consider the physics-based model developed by Chavas et al. (2015) that mathematically merges the Emanuel (2004) and the Emanuel and Rotunno (2011) solutions for the outer- and inner-core wind changes, respectively. However, these three models assume a perfect azimuthal symmetry structure of the wind fields (Fig. 1d,e,f), which can lead to large errors in storm surge forecasting (Xie et al., 2011). A more recent model, the Generalized Asymmetrical Holland Model (GAHM), is also

tested incorporating asymmetries (Fig. 1c) by considering information from all available isotachs in the quadrants (Gao et al., 2017; Dietrich et al., 2018; Bilskie et al., 2022).

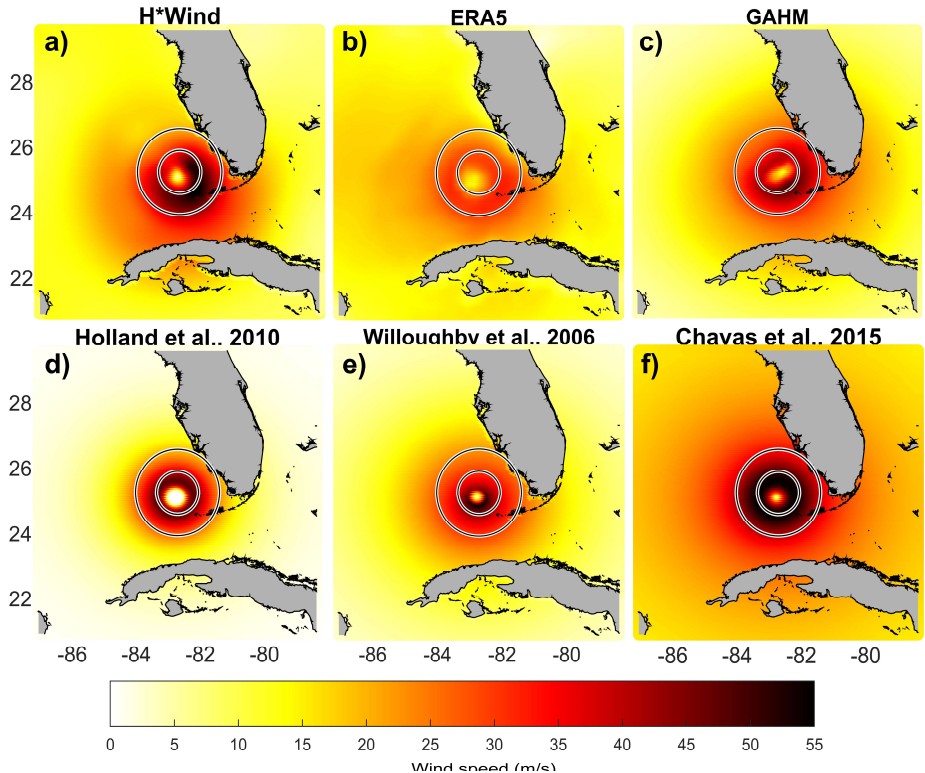

**Figure 1: Wind field in the different atmospheric forcings used in the study during hurricane Wilma before landfall in Florida (USA). a) H\*Wind real-time hurricane wind analysis system developed as part of the National Oceanic and Atmospheric**
**Administration (NOAA) Hurricane Research Division (Powell et al., 1998), considered here as the reference. b) ERA5 reanalysis data. c,d,e,f) The four different parametric wind models tested in the study.**

### 3.  Methods

Storm surges induced by four hurricanes are simulated in the northwestern Atlantic region using two different models (ADCIRC and NEMO) sharing similar configurations. These models are then used to conduct sensitivity experiments on the
atmospheric and ocean forcings. The simulated extreme storm surge events are evaluated in terms of maximum amplitude, duration, and correlation against tide gauge records.


### 3.1. Numerical models and regional configurations

The Advanced Circulation (ADCIRC) Model, here used in the v53 version, is a numerical model designed for simulating coastal hydrodynamics (Luettich et al., 1992; Westerink et al., 1994). It solves a formulation based on the Navier-Stokes equations for shallow water conditions, called the shallow-water equations. The equations are solved by discretizing spatial derivatives using a finite element method. This approach enables the use of unstructured meshes, which offers the advantage of high-resolution discretization in specific areas of interest—such as coastal regions or inland—without the computational cost of an increased resolution over the whole spatial domain. The model relies on the input of meteorological data on the ocean surface, specifically wind and pressure fields. The meteorological data can be sourced in different formats, including parametric wind models or gridded wind fields from reanalyses and climate models. Tidal levels or mean sea level forcing can be added as an optional input through the boundaries. The model has been mostly used in the 2DDI barotropic mode, resolving storm surges and tides, although it does offer the option to operate in a 3-D mode, which requires supplementary inputs such as temperature and salinity. Additional options can also include the modelling of the wetting and drying of inundated areas (Dietrich et al., 2004), the inclusion of river flows, the representation of obstructions to flow (Luettich and Westerink, 1999), and the integration of the wave setup by coupling with a wave model (Dietrich et al., 2012). ADCIRC has been widely used in research for storm surge modelling induced by tropical cyclones at various scales—global (Pringle et al., 2021), regional (Marsooli and Lin, 2018; Camelo et al., 2020; Gori et al., 2023), and more local (Yin et al., 2016; Dietrich et al., 2018). In addition, ADCIRC model is usually applied in emergency operational forecasting systems, such as the NOAA Operational Model (Riverside Technology, 2015). It is also utilized as the standard coastal storm surge model by the U.S. Army Corps of Engineers (USACE), and the U.S. Federal Emergency Management Agency (FEMA).

The ocean general circulation model NEMO (Nucleus for European Modelling of the Ocean) (Madec et al., 2023), is a numerical model designed for simulating the 3-D baroclinic ocean developed by a European consortium (https://www.nemo-ocean.eu/). It solves the primitive equations, i.e. the Navier-Stokes equations and a nonlinear equation of state that couples the temperature and salinity to the fluid velocity, with assumptions based on scale considerations. The equations are solved using a finite difference method. The ocean is discretized horizontally using a curvilinear ORCA grid, almost regular in our study area, and vertically using a chosen coordinate system, resulting in a high computational cost. The model relies on the input of atmospheric fields (air temperature, specific humidity, winds, atmospheric pressure, short- and longwave radiation, precipitation and snow cover) and tidal potential at the surface, oceanic fields (3-D ocean temperature, salinity, currents and 2-D sea level) at lateral boundaries in the case of a regional configuration, and river runoff fluxes. In addition to the storm surges and tides, NEMO can resolve the mean sea level i.e. ocean general circulation associated to baroclinic processes and addition of mass to the ocean. In this study, we only used the ocean circulation module of NEMO in its version 4.0.4 (Madec et al., 2019), which we refer to as NEMO but additional components can be included such as sea ice modelling and biogeochemical processes. NEMO has been recently used for sea level research at global (Royston et al., 2022) and regional scale (Adloff et al., 2018; Chaigneau et al., 2022). It is also used in the framework of the Copernicus Marine Service (CMEMS), providing free-of-charge ocean data and information derived from real-time systems and reanalyses at global and regional scales. For instance, it is utilized to forecast extreme coastal water levels and support coastal flood awareness applications at European scale (Irazoqui Apecechea et al., 2023).

ADCIRC and NEMO are intercompared for storm surge modelling in the northwestern Atlantic region. The domain extends from 98 to 55 ºW and from 6 to 31.5 ºN (Fig. 2). The region includes the whole Caribbean Sea, Gulf of Mexico and a part of the northwestern Atlantic Ocean. This region is prone to tropical cyclone development due to the warm water temperatures, moisture levels and wind patterns. A variety of oceanographic processes are found in this domain that are important to consider for storm surge modelling. The region contains strong variations of bathymetry, with a wide continental shelf in the Gulf of





Mexico and around the complex islands of Bahamas and a tight continental shelf around the Caribbean islands. In terms of ocean circulation, the dominant feature is the Gulf Stream that originates in the Gulf of Mexico and flows through the Straits of Florida (USA) and up the eastern coastline of the United States. Tidal amplitudes are relatively moderated in the region, with largest amplitudes reaching approximately 2 meters in the northern Surinam and Guyana, as well as in the northeastern Florida (USA). To ensure a fair comparison between both models, two similar configurations have been developed, operating in a 2-D barotropic mode. While it allows only storm surges and external tides to be resolved, a barotropic setup is expected to represent the main ocean response to tropical cyclones. As NEMO is mainly used in a 3-D baroclinic mode, the code has been modified to enable running the model in a barotropic mode for the region of interest. Modifications were implemented based on the Met Office configuration for the UK (O'Neill et al., 2016). The model operates using two vertical sigma levels with only one active layer and with typical baroclinic processes disabled. Tracers (temperature and salinity) remains constant in space and time so that changes in pressure gradient generating ocean circulations and transport are not considered. Vertical physics such as vertical mixing, internal waves, convection are entirely deactivated. Atmospheric inputs are restricted to winds (wind stress) and pressure (barotropic effects due to pressure forcing), with the total turbulent heat flux set to zero during the whole simulation. The resolution of the two configurations is limited by the computational cost of the NEMO model which has a quasi-regular resolution in the region, here of 9 km. A similar resolution has been chosen for the ADCIRC, spanning from 3 km near the coast or in shallow water areas to 70 km in the deeper open ocean (Fig. 2b). Both the bathymetry and the coastline taken from NOAA Operational Model with ADCIRC (Riverside Technology, 2015) have been interpolated on the ADCIRC and NEMO grids (Fig. 2a). In the NEMO configuration, dry areas are not allowed. Consequently, a minimum bathymetry value is set to 3 meters to allow lower sea levels, such as during low tides. The identical value is implemented for the ADCIRC configuration as well even if dry areas are allowed. Both configurations are driven by hourly winds and pressure from the ERA5 atmospheric reanalysis. In NEMO, the wind stress formulation has been updated to follow the same S&B scheme as in ADCIRC (Smith and Banke, 1975, eq. (1)). Additionally, they are forced by eight tidal constituents (M2, K2, S2, N2, Q1, O1, P1, K1) derived from TPXO9 (Egbert and Erofeeva, 2002). For each simulated hurricane, the model is also run with only the astronomical tidal forcing at the open boundaries, excluding meteorological forcing. This approach enables to isolate the storm surge component of the wind-driven simulations and compare to the non-tidal residuals from tide gauges. The description of the different settings and forcings used for ADCIRC and NEMO configurations are provided in the Appendix A (Tab. A1).

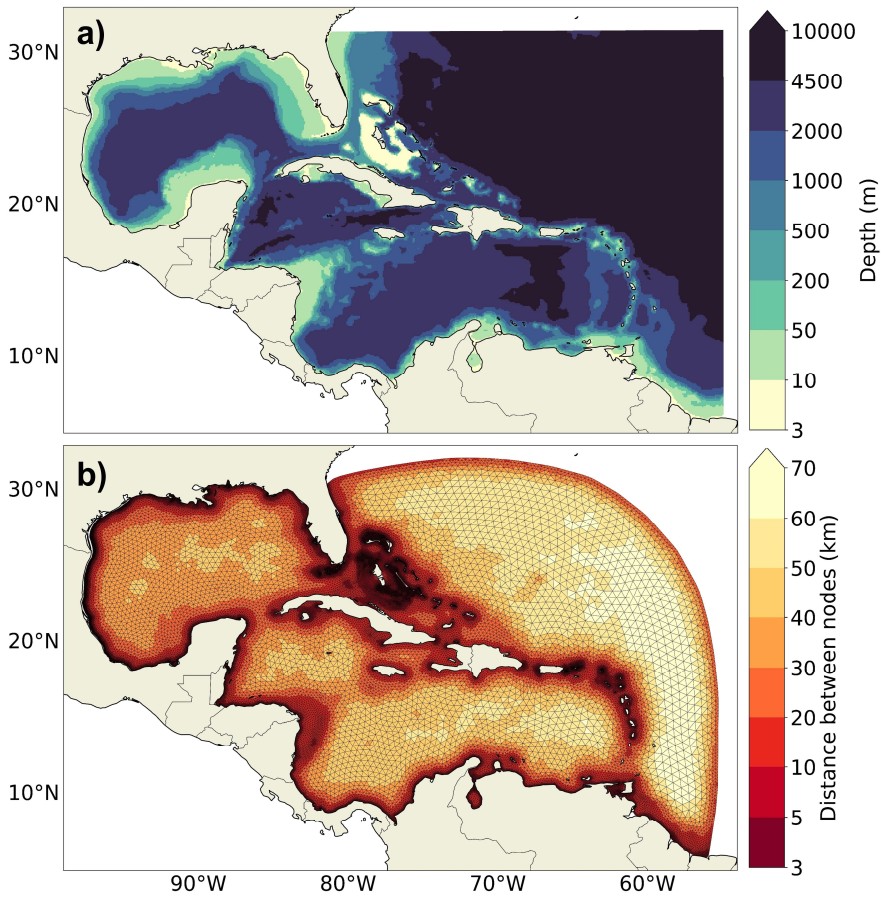


**Figure 2: a) Bathymetry used in the study and NEMO domain. b) ADCIRC domain and grid spacing with the ADCIRC unstructured mesh.**

### 3.2. Sensitivity experiments

Sensitivity experiments are conducted based on the developed configurations. Their aim is to assess the effect of atmospheric

and oceanic forcings, physical parameterizations on wind stress and baroclinic/barotropic modes on the performance of the

models in simulating storm surges. All the simulated experiments are listed in Table 3. The sensitivity of the atmospheric

forcing on storm surge modelling is assessed comparing the ERA5 atmospheric reanalysis and parametric wind models that

are usually applied for simulating tropical cyclones. This experience is conducted with the ADCIRC model as it is extensively

used and developed for this application, notably for operational systems (Fleming et al., 2008; Riverside Technology, 2015).

Then, the sensitivity of the ocean forcing on storm surge modelling is assessed in terms of non-linear interactions of surge with

the astronomical tide and variations in mean sea level. These experiments have been tested with the barotropic NEMO

configuration by introducing the daily mean sea level forcing from the GLORYS ocean reanalysis (Garric and Parent, 2017)

or by excluding tidal forcing at the boundaries. We have also conducted these tests using the ADCIRC model however we

chose to present only the experiment for NEMO as the results were consistent between both models. The barotropic

configuration of NEMO is also used to investigate the impact of wind stress parameterization on storm surges, thanks to the

flexibility of NEMO in modifying the code. Recent papers have highlighted a non-negligible impact of the selected

parameterization or of the tuning of the parameters utilized within it (O'Neill et al., 2016; Pineau-Guillou et al., 2020). The





S&B (Smith and Banke, 1975) scheme utilized in ADCIRC and applied in this study in NEMO (eq. (1)), is compared to the
Charnock formulation (Charnock, 1955) which is the reference formulation in NEMO (eq. (2)) :

$$S\&B: C_D = (0.75 + 0.067|U|)e^{-3} \text{ (1)}$$

$$\text{Charnock: } z_0 = \frac{\alpha u_*^2}{g} \text{ (2)}$$

with U the 10m wind speed, $z_0$ the bottom roughness, α the Charnock parameter, $u^2$ the friction velocity and g the gravity. In
the reference NEMO code, the parameter α remains constant in space and time, equal to 0.018. We also performed another
simulation with a variable Charnock parameter coming from ERA5 reanalysis outputs, thus depending on the waves (Riverside
Technology, 2015).

Finally, a sensitivity experiment on the importance of baroclinic motions on modelled storm surges is conducted, requiring the
utilization of a distinct configuration. This experiment is conducted with NEMO due to its standard operation in a baroclinic
mode. A baroclinic configuration has thus been set up based on Wilson et al., 2019. This configuration has 75 verticals levels
and is driven by the GLORYS ocean reanalysis (Garric and Parent, 2017) at the lateral oceanic boundaries and for initial state
with the variables described in the NEMO description part. In addition, it is forced by more atmospheric variables from ERA5
at the air-sea interface as explained in the NEMO description part. This configuration therefore allows changes in pressure
gradients (due to changes in temperature and salinity) generating ocean circulations and transport as well as vertical physics.
The short duration of the simulations (Tab. 1) does not allow the modelling of deep ocean circulation, but surface circulation
which occurs more rapidly can be simulated. The differences between the barotropic and baroclinic NEMO configurations are
summarized in the Appendix A (Tab. A1).

| Name of the experiment | Model | Type | Atmospheric forcing: winds and pressure | Tides | Other ocean forcing at boundaries | Wind stress formulation |
|---|---|---|---|---|---|---|
| ADCIRC_ERA5 | ADCIRC | 2D barotropic | ERA5 | Yes | No | S&B |
| NEMO_ERA5 | NEMO | 2D barotropic | ERA5 | Yes | No | S&B |
| ADCIRC_DHM, ADCIRC_Chavas, ADCIRC_Willoughby, ADCIRC_GAHM | ADCIRC | 2D barotropic | Parametric wind models: DHM, Chavas, Willoughby, GAHM | Yes | No | S&B |
| NEMO_msl | NEMO | 2D barotropic | ERA5 | Yes | Mean sea level (GLORYS, daily) | S&B |
| NEMO_without_tides | NEMO | 2D barotropic | ERA5 | No | No | S&B |
| NEMO_charnock | NEMO | 2D barotropic | ERA5 | Yes | No | Charnock: α=0.018 |
| NEMO_charnock_variable | NEMO | 2D barotropic | ERA5 | Yes | No | Charnock: α=variable |
| NEMO_baroclinic | NEMO | 3D baroclinic (75 levels) | ERA5 (+temperature, humidity, radiative fluxes, precipitations, snow cover) | Yes | GLORYS reanalysis: temperature, salinity, currents, sea level | Charnock: α=0.018 |

**Table 3: Sensitivity experiments performed with ADCIRC and NEMO models.**

### 3.3. Statistical evaluation of the extreme events

Hourly outputs from all the simulations performed (Tab. 3) are statistically compared to tide gauge records during the four
hurricane events. We have developed an automated method to identify the time window of each storm surge extreme event in
order to evaluate not just the maximum storm surge reached during each hurricane but also its temporal behavior. The time
       window is identified at each tide gauge station for each hurricane. The storm surges from the various simulations are extracted
       at the closest point to each tide gauge station, and the same time window is applied to each. To select the appropriate time
       window for each extreme event in the tide gauge records, we found that the wind time series closest to each tide gauge location
       was a good indicator of the storm conditions. The following steps are applied to extract the time window: for each ERA5 wind
time series under the hurricane simulation dates (Tab. 1), the maximum and local maxima are identified as well as the inflection
       points on either side of the maximum wind speed. The selected time window to identify the extreme event is defined by the
       two inflexion points that include the maximum wind speed and all local maxima exceeding the 95th percentile threshold, as
       illustrated in Figure 3.

       Once the time window has been defined, different statistical metrics are applied to validate the modelled storm surge against
tide gauge data. First, the maximum values reached within the specified time window are compared between the simulations
       and tide gauge data. The mean absolute error (MAE) is used to derive a general skill value for all the selected tide gauges. The
       second step is to evaluate the storm surge time series. This is done by computing the Pearson correlation coefficient and the
       difference in duration, in hours, within the extreme event, exceeding the 90th percentile of the storm surge time series. In
       addition to the maximum value reached, these metrics depending on the temporal behavior can be also important for impact
assessments (e.g. accelerated coastal erosion and increased likelihood of coastal flooding).

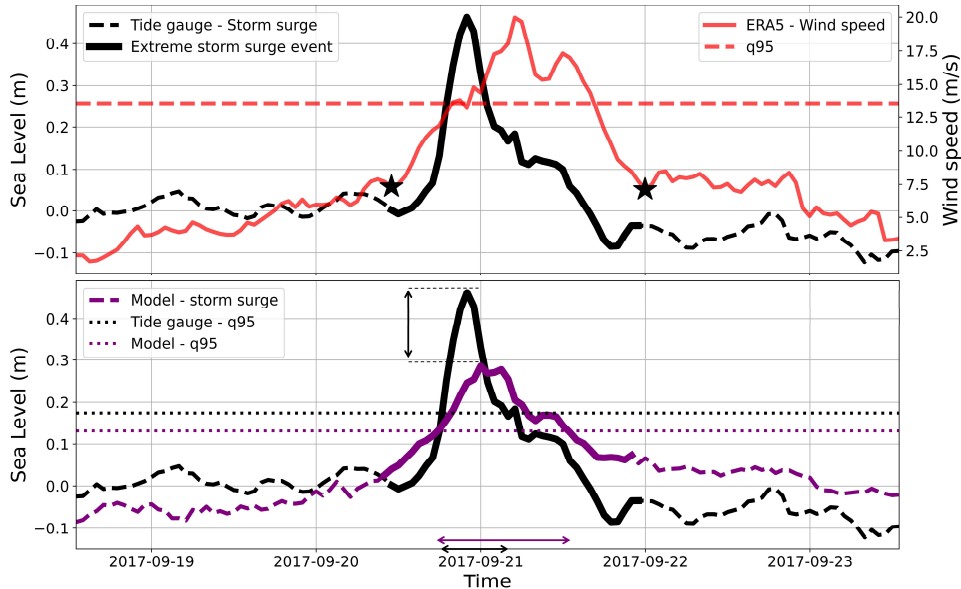

**Figure 3:** Sketch of the selection of the storm surge extreme event for tide gauge records
(top) and comparison with simulations (bottom). The storm surge data extracted from the GESLA3 dataset at one station is
represented in black. The wind speed obtained from ERA5 is shown in red, with the dashed line denoting the 95th percentile
threshold. The simulated storm surge data is represented in purple. The two stars and wider black and purple lines indicate the
beginning and the end of the time window defining the storm surge extreme event. The vertical black arrow denotes the difference
in the observed and modelled maximum storm surge reached within the time window. The horizontal black and purple arrows
denote the duration above the 90th percentile for both observed and modelled data.





## 4. Results

### 4.1. Inter-model comparison


ADCIRC and NEMO simulations are intercompared for storm surge modelling using two similar configurations and the same storm surge drivers. The comparison between storm surge maximum generated by ADCIRC forced by ERA5 and tide gauge data is presented for the four hurricanes (Fig. 4). The validation is restricted to a few points (Tab. 2) due to the scarcity of tide gauge data along the coasts of Cuba, Haiti, and northern Mexico, i.e. areas significantly impacted by three of the four hurricanes

(Tab. 1). The overall spatial pattern of the modelled storm surges appears consistent with the tracks of the hurricanes. Both, observed and modelled highest storm surges, exceed one meter for each hurricane, however, ADCIRC simulations tends to underestimate the maximum compared to tide gauge data, especially along the eastern coast of Florida (USA) and in the Caribbean Islands (Fig. 4).

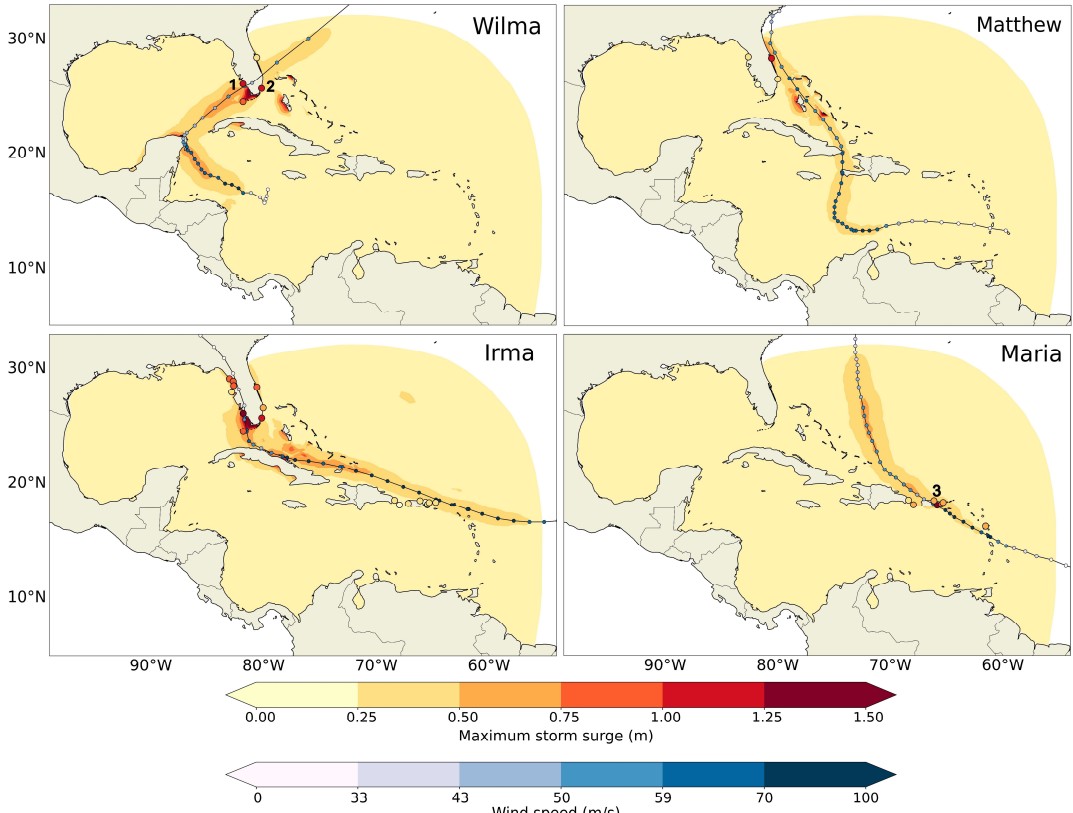

**Figure 4: Modelled (ADCIRC with ERA5 forcing, map) and observed (tide gauges, circles) maximum storm surge for the four simulated hurricanes. The tracks of the hurricanes and wind speed are shown in blue. The blue colorbar represents the different hurricane categories, from category 1 between 33 and 43m/s to category 5 for winds higher than 70m/s. The locations used to analyze time series are marked with numbers 1 (Naples_FL), 2 (Virginia_Key_FL) and 3 (San_Juan_PR).**

Time series of different target locations are shown in Figure 5 to analyze the response of both ADCIRC and NEMO models.

The models exhibit very similar responses, with occasional instances where one model outperforms the other. For instance, NEMO displays slightly better correlation for Wilma at Naples station during the post-peak period, while ADCIRC performs better in capturing the surge amplitude for Irma at Virginia Key station. In general, the correlation between the models and observed data is well reproduced. Along the western coast of Florida (Naples), both models also satisfactorily simulate the surge amplitude including the double-peak behavior during hurricane Wilma. However, for all simulated hurricanes, the storm




290 surge is notably underestimated by both NEMO and ADCIRC along the eastern coast of Florida (Virginia Key) and in the

Caribbean Islands (San Juan) (Fig. 5) as also illustrated in Figure 4.

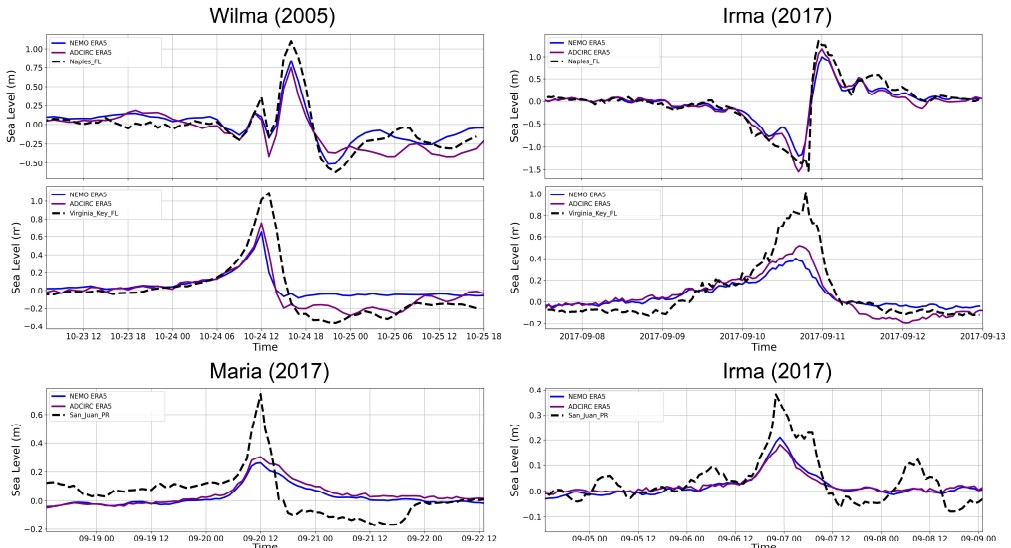

**Figure 5: Modelled (blue and purple lines) and observed (black dashed line) storm surge time series at three tide gauge locations, in the Florida region (top, center) and in the Caribbean region (bottom). The locations are marked in Figure 4. Results are shown for**
295 **hurricanes Wilma, Irma and Maria (Tab. 1).**

The similarity of the results between ADCIRC and NEMO is noticeable when considering all tide gauges available as well

(Fig. 6), revealing a general underestimation, with NEMO showing slight improvements for hurricane Matthew. In comparison

to others studies at this scale and resolution, both models demonstrate satisfactory performance for three of the four hurricanes

with a mean absolute error of less than 0.3 m (Muis et al., 2019; Dullaart et al., 2020; Muis et al., 2020; Wood et al., 2023).

300 However, both ADCIRC and NEMO notably underestimate storm surges associated with hurricane Maria, located north of the

Caribbean Sea (Fig. 6). The time series in the Caribbean (San Juan) consistently show significant underestimations for both

hurricanes Irma and Maria (Fig. 5). It suggests a region-dependent underestimation rather than one dependent on the hurricane

characteristics. For hurricane Irma, in contrast to Maria, other tide gauges are utilized for statistical analysis, particularly along

the western coast of Florida where models perform well (Fig. 5), resulting in an overall good performance (Fig. 6).

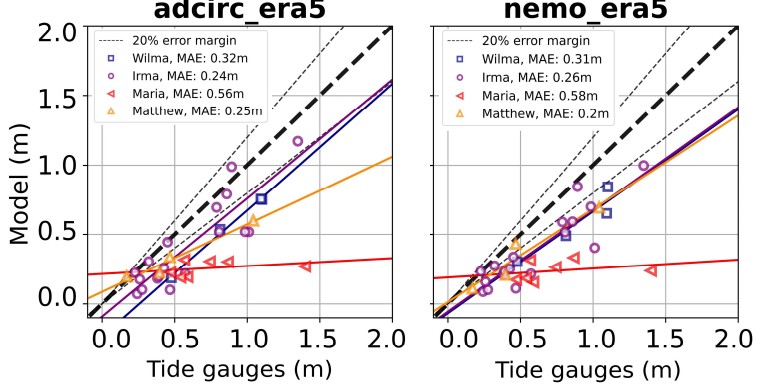


**Figure 6: Scatter plot of the modelled vs observed maximum storm surge for the four hurricanes for ADCIRC (left) and NEMO (right) simulations. The MAE value represents the mean absolute error on the surge maximum.**




The general correlation is also consistent between ADCIRC and NEMO, with a satisfactory mean value of over 0.8 for both models (Fig. 7), aligning with recent literature (Muis et al., 2019; Dullaart et al., 2020; Muis et al., 2020). The extreme event

duration above a high percentile is also presented as it combines both the biases on the surge amplitude and correlation. Hurricanes Wilma and Irma show a slight underestimation of the extreme event duration around 25% for both models compared to tide gauge data (Fig. 7). This doubles to 50% for hurricanes Maria and Matthew. In the case of hurricane Maria, although the correlation is high, the surge levels are significantly underestimated. Analysis of ERA5 meteorological inputs (not shown) indicates accurate hurricane track representation but notable biases in meteorological conditions, particularly

around the Caribbean islands. In particular, we observed weaker extreme winds and higher atmospheric pressure in the eye of the hurricane. These biases might be attributed to factors such as reduced data assimilation in this region or the impact of the resolution of the reanalysis (i.e. difference in the land mask of ERA5 over the Caribbean islands, with about 0.25º of spatial resolution). For hurricane Matthew, the underestimation of the event duration is due to a lack of correlation with observations attributed to the hurricane track being farther from the coast and tide gauges compared to other hurricanes.

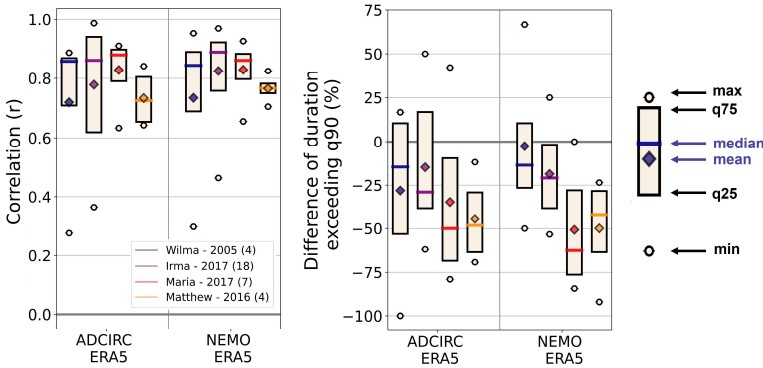


**Figure 7: Boxplot of the correlation (left) and the difference in the storm surge duration above the 90ᵗʰ percentile (right) for ADCIRC and NEMO simulations. The number of tide gauges considered for each box is in brackets after the four hurricane names.**

**4.2. Analysis of the sensitivity experiments on storm surge modelling**

Given the similar performance of ADCIRC and NEMO models, we are employing each model strength to conduct sensitivity

experiments (Tab. 3). These experiments aim to assess the effect of atmospheric and oceanic forcings, physical parameterizations on wind stress and baroclinic/barotropic modes on the performance of numerical models in simulating storm surges.

First, we assess the impact of the atmospheric forcing by comparing storm surges using the four parametric wind models to those simulated using ERA5. The comparisons shown in Figure 8 are performed at the same three tide gauge locations as in

Figure 5. The results with parametric wind models are highly variable, displaying a range of performance between good and significant under- or over-estimates, which is very dependent on the location relative to the track of the cyclone. In general, the maximum surge values obtained through parametric wind models appear less satisfactory than those derived from the wind fields of ERA5 reanalysis. The parametric wind models rather accurately capture the peaks at Naples station, due to the close proximity of both hurricanes Wilma and Irma to the tide gauge (within 25 km). However, these peaks exhibit a time lag with

the parametric models compared to both ERA5 and tide gauges during hurricane Wilma. This is due to differences in the location of the hurricane track between the best track data and ERA5. According to the best track data, Wilma passes slightly earlier and closer to the Naples station (not shown). The behavior of the axisymmetric wind models also depends on the relative distance to the track. During hurricane Wilma, a significant fall in the storm surge is detected at Virginia Key station because the wind direction pushes the water away from the shore. At the same station, during hurricane Irma, the peak is significantly



underestimated because the center of the hurricane is further away (more than 100 km). The use of the GAHM model shows significant differences compared to the other three models, occasionally improving the maximum surge, as at Virginia Key station, and sometimes worsening it, as at Naples station, probably due to the consideration of asymmetries (Fig. 1c). Nevertheless, a notable improvement in the performance of parametric wind models is observed for hurricane Maria (Fig. 8 and 9a), where the maximum surge was highly underestimated using ERA5 forcing, highlighting the relevance of the use of

parametric winds in such cases.

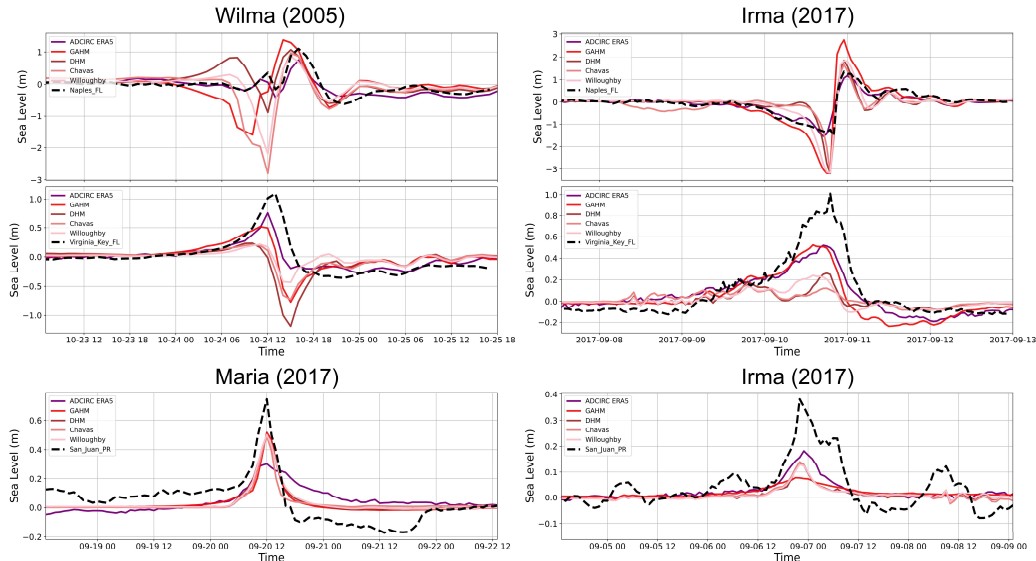

**Figure 8: Modelled (color lines) and observed (black dashed line) storm surge time series at three tide gauge locations, in the Florida region (top, center) and in the Caribbean region (bottom). Each color represents a simulation with a different atmospheric forcing (ERA5 or a parametric wind model). The locations are marked in Figure 4. Results are shown for Wilma, Irma and Maria hurricanes**
**(Tab. 1).**

   The correlation between parametric models and the observations falls below 0.6 in average, varying significantly among hurricanes (Fig. 9). For instance, the DHM, Willoughby and Chavas models perform rather poorly for hurricane Wilma, while the GAHM model does so for hurricane Matthew. These differences in correlation compared to ERA5 atmospheric forcings are most likely due to the simplifications used to generate atmospheric surface conditions in the parametric models (Fig. 1), in

contrast to the complete wind spatial fields considered when using ERA5. As explained above, it is also due to the location of the hurricane track in the parametric models and its relative distance to the tide gauges. For all simulated hurricanes, the duration of extreme events is consistently underestimated across all simulations (Fig. 9). While ERA5 systematically underestimates less than 50% of the time, parametric wind models tend to exhibit more substantial underestimations and even miss some extreme events.


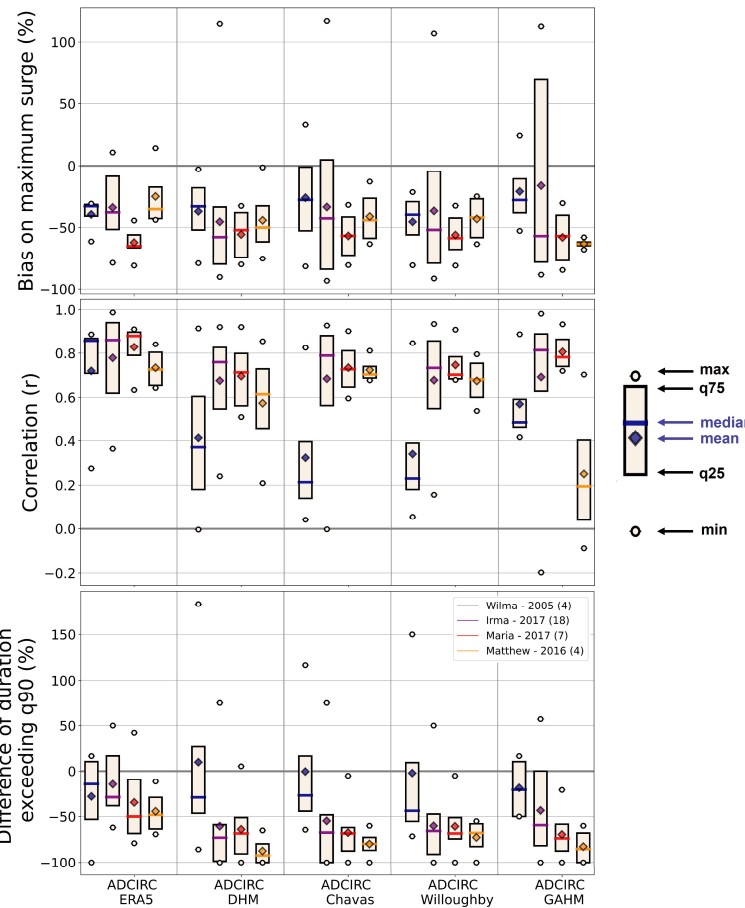

**Figure 9: Boxplot of the maximum surge bias (top), correlation (middle) and the difference in the duration of the surge above the 90th percentile (bottom) for the simulations using different atmospheric forcings. The number of tide gauges considered for each box is in brackets after the four hurricane names and dates. A difference of 100% corresponds to a missed extreme event.**

Then, we assess the effect on storm surge due to non-linear interactions with the astronomical tide and variations in mean sea

level, as well as the sensitivity to different wind stress schemes. In addition, the baroclinic contribution to storm surges is also studied using a 3-D configuration with 75 vertical levels. Storm surges simulated by NEMO for the different aforementioned experiments are compared to tide gauge data at three locations in Figure 10. Non-linear interactions of tides and mean sea level with storm surges have minimal contribution to the extreme sea levels, as well as the different experiments on the wind stress formulation. The baroclinic response however significantly improves by up to 40 cm the maximum storm surge estimates at

Virginia Key station in the southeastern Florida peninsula and also slightly in the Caribbean.

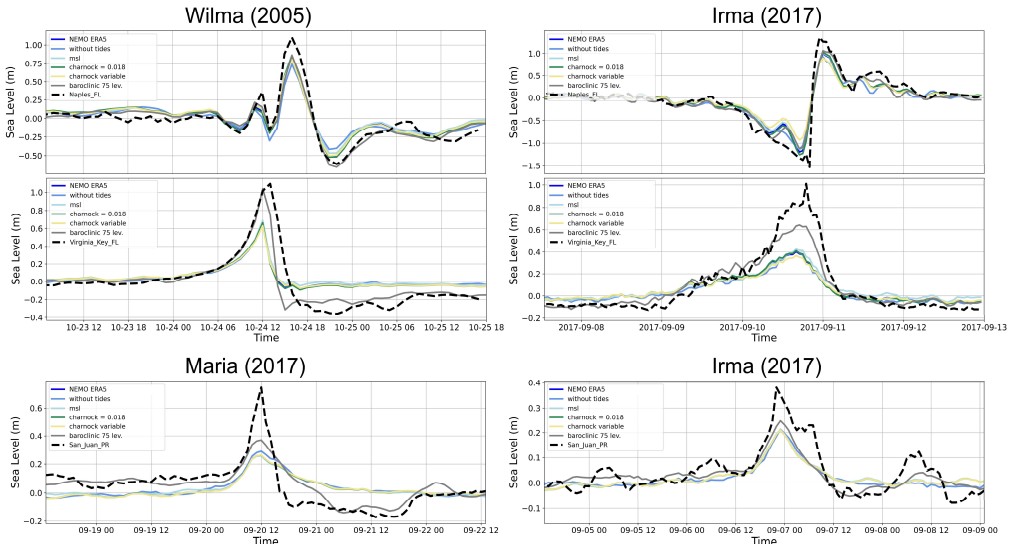

**Figure 10: Modelled (color lines) and observed (black dashed line) storm surge time series at three tide gauge locations, in the Florida region (top, center) and in the Caribbean region (bottom). Each color represents a different experiment (accounting for different sea level processes or wind stress implementation). The locations are marked in Figure 4. Results are shown for Wilma, Irma and**
**Maria hurricanes (Tab. 1).**

In general, for all the simulated hurricanes, the baroclinicity significantly influences the maximum surge amplitudes, reducing the bias by approximately 10 to 20% in average, depending on the hurricane (Fig. 11). Nevertheless, the average correlation of the surge peak events remains virtually no impacted by the baroclinic experiment. This is due to compensations between minor improvements such as during the post-storm periods for hurricanes Wilma and Maria at Naples and San Juan stations,
and degradations observed at Virginia Key during the decreasing surge for Wilma. The statistics for the other experiments are not presented as their impact on the surges is very small. The baroclinic impact is also substantial for event duration with less than 20% of underestimation for all the simulated hurricanes.

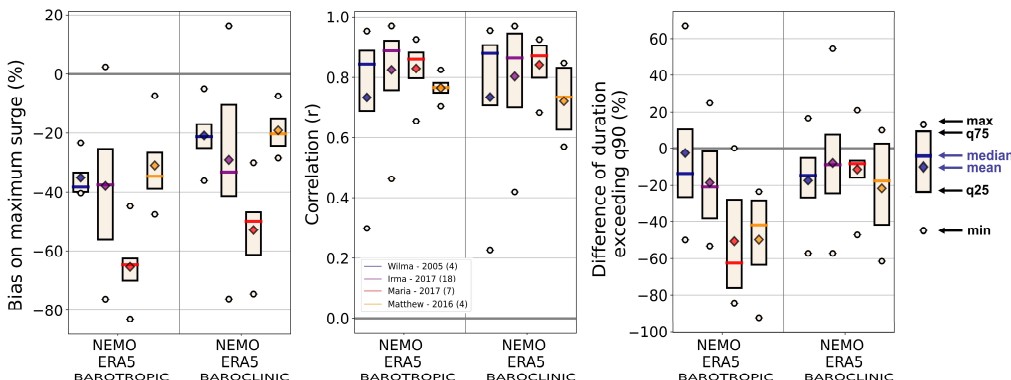

**Figure 11: Boxplot of the maximum surge bias (left), correlation (middle) and the difference in the duration of the surge above the**
**90th percentile (right) for the NEMO barotropic and baroclinic simulations. The number of tide gauges considered for each box is in brackets after the four hurricane names and dates.**

Comparing the maximum surges in the baroclinic and barotropic simulations provides insights into the regional significance and locations of the baroclinic impact (Fig. 12). Substantial differences of more than 20 cm are identified along the eastern coast of Florida when the hurricane is coming from the eastern domain, and various tide gauges were utilized in that zone (Fig.

4). In other areas, such as the Caribbean for hurricane Maria or the Yucatan Peninsula (Mexico) for hurricane Wilma, differences of more than ten centimeters are observed along the hurricane track in coastal areas.

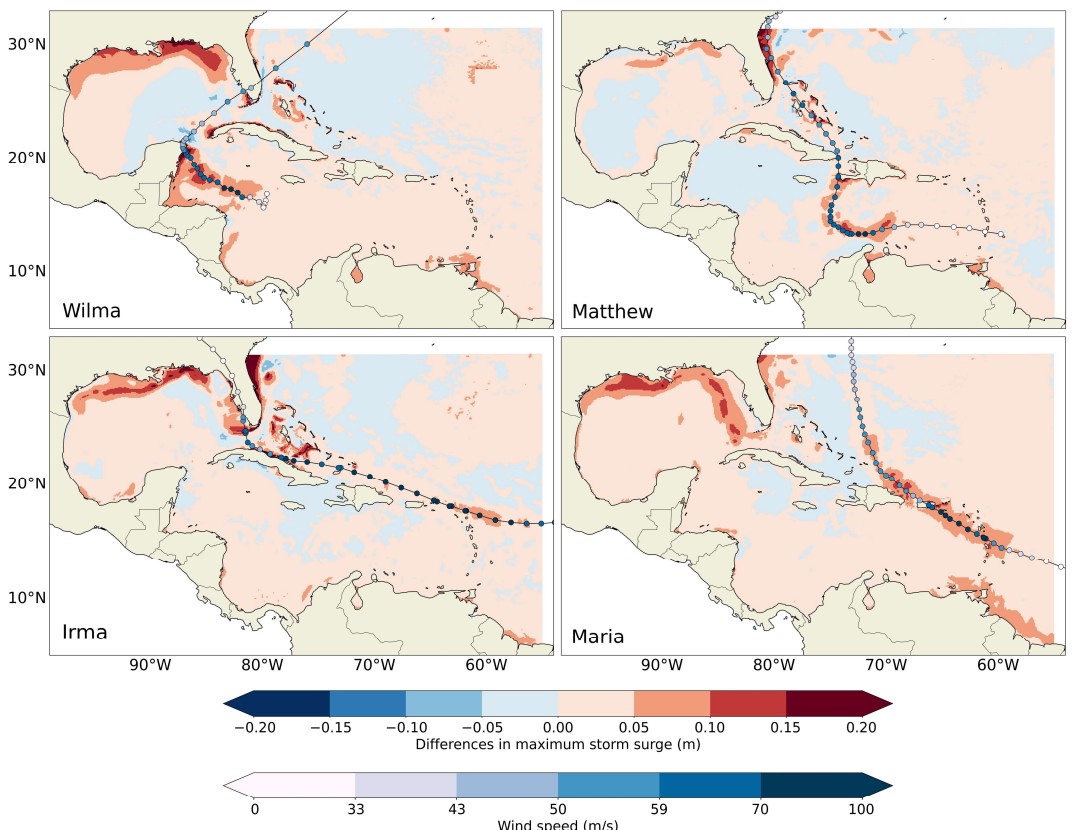

**Figure 12: Differences in storm surge maximum between baroclinic and barotropic simulations. Results are shown for the four simulated hurricanes Wilma, Matthew, Irma and Maria (Tab. 1). The tracks of the hurricanes and wind speed are shown in blue.**
**The blue colorbar indicates the different hurricane categories, from category 1 between 33 and 43m/s to category 5 for winds higher than 70m/s.**

The passage of the hurricane affects the general surface circulation, for instance through winds inducing mixing in the water column, leading to a decrease in sea surface temperature. In addition, the baroclinic simulation considers other atmospheric variables associated with hurricanes, such as precipitations. It does not only impact the local sea level budget but also modify

the circulation due to the effect on salinity, interacting with the existing circulation. As a result, each cyclone generates a distinct baroclinic response, influenced by its specific characteristics and by interactions with the local oceanographic features. Other studies have investigated the impact of baroclinic motions on storm surges. For example, Ye et al., 2020  implemented a regional 3-D baroclinic model, comparing it to a 2-D barotropic model in simulating storm surges induced by hurricanes along the US east coast. The results revealed a non-negligible influence of baroclinicity during the post-storm period, with

differences of up to 14% in sea level amplitude. However, this study focused on a single hurricane, and comparisons to observational data were conducted in an estuarine area outside our domain. Another study by Ezer, 2018 examined interactions between hurricane Matthew, the Gulf Stream, and coastal water levels using a more basic model. The study highlights an increase in storm surge along the eastern coast of Florida with a similar order of magnitude to our findings. It is attributed to the passage of the hurricane reducing the sea surface height slope between the coast and the other side of the Gulf Stream,

consequently reducing the geostrophic Gulf Stream flux and increasing the sea level on the coast.


## 5. Discussion

Results obtained for the inter-model comparison between ADCIRC and NEMO and for the different sensitivity experiments are summarized and discussed in this section. To that aim, two synthesis figures are provided: the variance decomposition of the different sources of uncertainty (Fig. 13) and the mean absolute error (MAE) on the maximum storm surge (Fig. 14). For

the variance decomposition, we have decomposed the total ensemble uncertainty from the different sources of uncertainty and the interactions among them using a n-factor ANOVA-based variance partition method (Storch and Zwiers, 1999). The sources of uncertainty considered include: the selection of simulated hurricanes, numerical models, atmospheric forcings, ocean forcings, physical parameterizations, and barotropic/baroclinic modes.

When used with a similar configuration (domain, resolution, bathymetry, barotropic), ADCIRC and NEMO simulate storm

surges due to tropical cyclones in a similar way compared to tide gauges regardless of the simulated cyclone. This positive outcome highlights the potential of NEMO, which is currently rather poorly employed for this application. This result is illustrated by the variance decomposition (ANOVA) depicted in Figure 13a, where the variability of the three different metrics (maximum value, bias on maximum value, and correlation) is only dependent on the simulated hurricanes and not on the chosen numerical model.

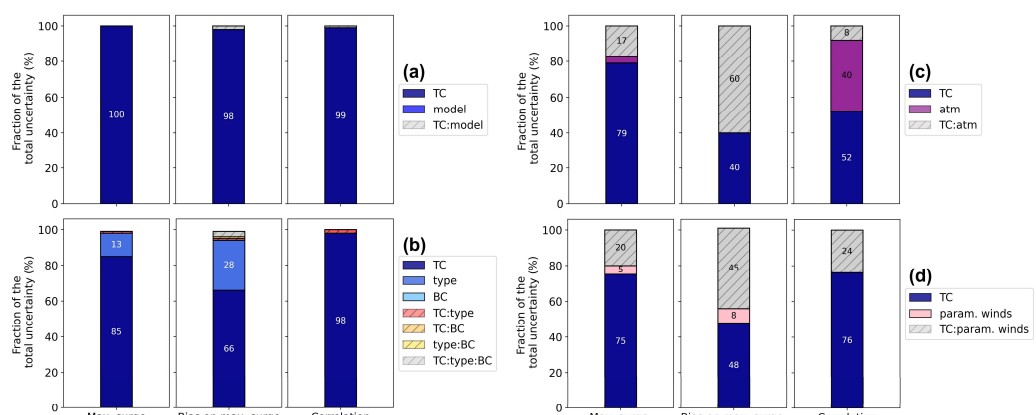


**Figure 13: Relative contribution (between 0 and 100%) of different sources of uncertainty in the skill to model storm surges (maximum surge, bias on the maximum surge and correlation), based on two (a,c,d) and three-factor (b) ANOVA decomposition. To ensure a consistent number of values between the different sources of uncertainty, the four tide gauge locations where the highest surges occur are selected for each hurricane. Fraction of variance due to the four tropical cyclones (TC, dark blue) and (a) the two**

**models ADCIRC and NEMO (model, blue) for a total of n=16 values, (b) the model type i.e. baroclinic or barotropic (type, blue) and the boundary conditions i.e. with or without tides (BC, light blue) for a total of n=64 values, (c) the atmosphere i.e. ERA5 or Dynamic Holland Model (atm, purple) for a total of n=16 values, (d) the parametric wind models i.e. Dynamic Holland Model, GAHM, Willoughby, Chavas (param. winds, pink) for a total of n=64 values. Interactions between the different sources of uncertainties are noted with dashed lines. Experiments (c) and (d) are performed with ADCIRC and (b) with NEMO. The ANOVA**

**decomposition has been performed using "statsmodels" Python package.**

The performance of these models is however significantly impacted by those of the atmospheric reanalysis forcing hence varying across regions. In general, both models underestimate the storm surge amplitudes, which is a common feature in modelling studies at large scale (Kirezci et al., 2020; Irazoqui Apecechea et al., 2023). It is often associated with meteorological forcing issues, such as too weak extreme winds in the models and biases in capturing the tracks of the tropical cyclones (Hodges

et al., 2017; Dullaart et al., 2020; Gori et al., 2023). In our case, the use of the global ERA5 atmospheric reanalysis with a 1/4° resolution contributes to an inadequate resolution of the atmospheric processes particularly in complex land features, such as for hurricane Maria in the Caribbean islands. Additionally, the storm surge performance skill in the models is influenced by the quantity of assimilated data in the atmospheric reanalysis, which varies depending on the location (Hersbach et al., 2020). In our tropical Atlantic region, using ERA5 to simulate the storm surge amplitudes generally outperforms various parametric

wind models, except for hurricane Maria, as shown with the mean absolute error on maximum surge in Figure 14c. In addition,



as illustrated with the ANOVA in Figure 13b, the variance of the maximum surge depends mostly on the simulated hurricane (i.e., on the location) rather than on the atmospheric forcing. This means that in regions with less accurate reanalysis data, parametric wind models might serve as an alternative. For instance in Wood et al., 2023, the Dynamic Holland Model performs better compared to ERA5 in the southern China Sea, where ERA5 struggled in accurately capturing typhoon dynamics,

probably due to the smaller amount of assimilated data. However, in terms of correlation and duration of extreme events, using ERA5 to model storm surges significantly outperforms the various parametric wind models. This trend is consistent across all simulated hurricanes which is illustrated by a large part of the variance dominated by the atmospheric forcing in Figure 13c. When intercomparing parametric wind models, none appears superior since their performance highly depends on the hurricane being simulated and only four are simulated. This is highlighted by the large interactions between tropical cyclones and

parametric winds in the variance analysis on Figure 13d. An alternative approach could involve a combination of reanalysis and parametric models based on the specific region or the prevalence of tropical cyclones to consider the strength of each approach (Dullaart et al., 2021). To mitigate biases associated with meteorological forcing, potential strategies could involve the application of bias correction techniques (Li et al., 2019; Lemos et al., 2020) or the use of statistical or dynamical downscaling at higher resolutions to capture processes that are not resolved in global or regional reanalyses (Dullaart et al.,

2024). In recent developments, data-driven techniques, such as those employed by Tadesse et al., 2020 and Qin et al., 2023, utilize satellite products to quantify the relationships between storm surges and key atmospheric variables like wind speed and mean sea level pressure. These diverse methodologies offer a range of options for improving storm surge modelling accuracy and addressing region-specific challenges.

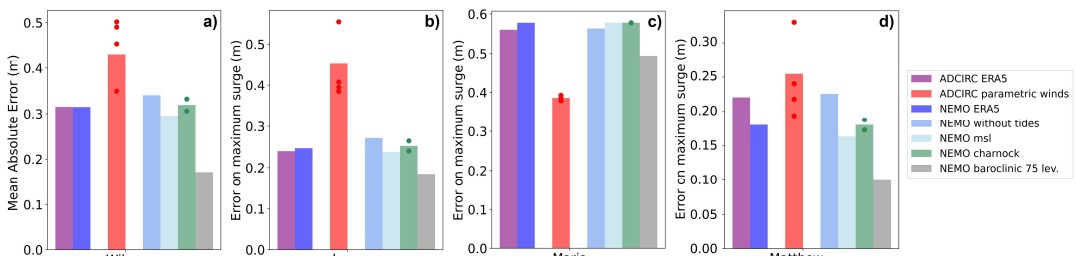

**Figure 14: Mean absolute error on the maximum surge values for all the different experiments performed (Tab. 3). The dots represent the errors of each experiment when they are grouped in a bar: for the parametric wind models (DHM, Chavas, Willoughby, GAHM) and for wind stress (constant and variable charnock parameters).**

Non-linear interactions of tides and mean sea level with storm surges as well as different wind stress formulations have shown a minimal impact on the storm surge estimates induced by hurricane events (Fig 13b, Fig. 14). The limited tide-surge

interaction simulated could be attributed to the small tidal range within the domain, rarely exceeding 2 meters. The impact of interactions with tides would be probably larger with a higher resolution in the coastal regions (Hsiao et al., 2019) or considering other regions dominated by tides, such as in the English Channel (Fernández-Montblanc et al., 2019; Arns et al., 2020), or in Asia with typhoons (Idier et al., 2019; Hsiao et al., 2019). Although the effect of mean sea level forcing appears small, its significance may become more pronounced in a long-term context, particularly when accounting for mean sea level

rise (Fox-Kemper et al., 2021). The inclusion of wetting and drying in NEMO, not employed in this study, would also contribute to better resolving ocean dynamics in shallow water areas, where storm surges are the biggest (O'Dea et al., 2020). In general, improving storm surge estimates by resolving more relevant components and their interactions may require the incorporation of additional processes and the use of a higher resolution model, together with a refined coastline and bathymetry. For instance, improving storm surge modelling could involve a coupling with a wave model to simulate wave setup and

associated interactions, as existing for ADCIRC with the SWAN wave model (Marsooli and Lin, 2018; Dietrich et al., 2018; Hsu et al., 2023) or for NEMO with WAM (Staneva et al., 2021) or WW3 wave model (Couvelard et al., 2020). Additionally, the consideration of river inflows would also be important to include the significant precipitation associated with hurricane


passages. To go further, employing a fully coupled ocean-atmosphere-waves model, or alternatively, a simpler atmospheric boundary layer model with a lower computational cost, would enable the simulation of the processes and feedback mechanisms

between the ocean and atmosphere (Lemarié et al., 2021). When coupled with increased resolution, adjustments based on land cover data, including the use of the Manning coefficient, canopy coefficient to mitigate wind stress from vegetation, directional effective roughness length could also be applied to refine surge amplitudes at the coast (Dietrich et al., 2018).

The inclusion of the baroclinic response have notably impacted storm surge amplitudes for all hurricanes (Fig. 13b) with less underestimated amplitudes as shown in Figure 14 with smaller MAE values. For instance, the model underestimations of the

maximum are reduced by up to 40 cm for a station in the southeastern Florida peninsula (USA). The correlation however remains unchanged as shown with the negligible contribution of baroclinic simulation to the variance for this metric in Figure 13b. These improvements are attributed to large changes in the general surface circulation caused by the hurricane passage. It is however important to note that these simulations are computationally very expensive (around 70 times longer), posing challenges for large scale studies or long-term applications, such as global to regional hindcasts or projections, but also for

operational purposes requiring efficient results. Alternative methods could be considered to incorporate these baroclinic processes without simulating the entire 3-D column, such as using the models in a 2-D baroclinic mode (Westerink and Pringle, 2018) or adding the baroclinic contribution as a post-processing step (Zhai et al., 2019).

## 6.   Conclusions

This study aimed to explore multiple factors affecting the performance of the numerical modelling of extreme sea levels

dominated by large storm surges induced by hurricanes. The factors explored encompassed the numerical model (ADCIRC and NEMO), the choice of the oceanic and atmospheric forcings, physical parameterizations for wind stress, and baroclinic/barotropic modes. Four historical hurricanes were simulated in the tropical Atlantic region, covering the Caribbean Sea and Gulf of Mexico. The modelled storm surge maxima and the behavior of the hourly time series were evaluated against tide gauge data.

The analysis of the different numerical experiments revealed some interesting insights. Both ADCIRC and NEMO numerical models can simulate storm surges due to tropical cyclones in a similar way compared to tide gauges. The performance of these models is however highly dependent of those of the atmospheric forcing hence varying across regions. In the analyzed tropical Atlantic region, the ERA5 atmospheric reanalysis forcing generally outperforms the various parametric wind models for storm surge modeling, in terms of maximum values, correlation, and duration of extreme events. The inclusion of the baroclinic

response significantly improves storm surge amplitudes, i.e. significantly reduces underestimates, in some regions such as along the southeastern Florida peninsula (USA). However, non-linear interactions of tides and mean sea level with storm surges as well as different wind stress implementations show very small contribution to the storm surges induced by hurricanes. These methodological insights can have key implications for the development of hindcast and projections, but also for coastal impact assessment, particularly for understanding and predicting hurricane-induced coastal flooding.

These results primarily rely on the available tide gauge data, which are scarce and occasionally out of service during hurricane events. Currently, there is a lack of alternative observational products to accurately measure storm surges, and satellite data remains insufficient for capturing local surge details near the coast (Lobeto and Menendez, 2024).

## Appendix A

| Model | ADCIRC | NEMO | NEMO baroclinic |
|---|---|---|---|
| Version | v53 | v4.0.4 | v4.0.4 |
| Resolution | From 3 km to 70 km | 1/12º (~ 9 km) | 1/12º (~ 9 km) |





| Type | 2-D Barotropic | 2-D Barotropic | 3-D Baroclinic |
|---|---|---|---|
| Number vertical levels | 1 | 2 (only one active) | 75 |
| Time step | 18 s | 18 s (barotropic motions), 600 s (baroclinic time step) | 18 s (barotropic motions), 600 s (baroclinic time step) |
| Time of calculation for 1 tropical cyclone | ~20 min | ~35 min (1 node) | ~2.5 h (15 nodes) |
| Vertical coordinates | sigma | sigma | z levels (partial steps) |
| Bathymetry and coastline | NOAA Operational Model with ADCIRC | NOAA Operational Model with ADCIRC interpolated on curvilinear 1/12 ° grid | NOAA Operational Model with ADCIRC interpolated on curvilinear 1/12 ° grid |
| Minimum bathymetry | 3 meters | 3 meters | 3 meters |
| Bottom stress | Quadratic friction (constant drag=2.5e-3) | Quadratic friction (constant drag=2.5e-3) | Quadratic friction (constant drag=2.5e-3) |
| Atmospheric forcing: | ERA5: hourly winds and pressure | ERA5: hourly winds and pressure | ERA5: hourly winds, pressure, temperature and specific humidity, radiative fluxes, precipitation, snow cover |
| Wind stress | S&B scheme: Cd=(0.75+0.067U)e-3 | S&B scheme: Cd=(0.75+0.067U)e-3 | Charnock (alpha=0.018) |
| Lateral boundary forcing: Ocean | no | constant tracers | GLORYS (1/4 °), daily tracers (temperature and salinity), currents, sea level |
| Lateral boundary forcing: Tides | 8 primary constituents TPXO9 | 8 primary constituents TPXO9 | 8 primary constituents TPXO9 |
| Initial conditions: | no | constant tracers | GLORYS (1/4 °) temperature and salinity of the day before the hurricane |
| Runoff | no | no | no (but the impact on the tracers is accounted for) |
| Sea level accounted for | Tides, storm surges | Tides, storm surges | Tides, storm surges, mean sea level (due to oceans circulations and variations in sea level budget) |

**Table A1: Table of the different configurations developed and settings used in them.**

## 520 Code availability

The NEMO 4.0 version used was developed by the NEMO consortium (https://doi.org/10.5281/zenodo.3878122, Madec et al., 2019). All specificities included in the NEMO code are freely available (NEMO, 2024: https://www.nemo-ocean.eu/). The ADCIRC code is available from the project website (http://adcirc.org/) under the terms stipulated there and is free for research or educational purposes.

## 525 Data availability

The tide gauge data used for validation are available on the GESLA website (at www.gesla.org). ERA5 atmospheric forcings are available on the Climate Data Store (https://cds.climate.copernicus.eu) in the context of the Copernicus Climate Change Service (C3S). The tidal forcing is available on the OSU TPXO Tide Models website (https://www.tpxo.net/home). The GLORYS ocean reanalysis product was obtained from the Copernicus Marine Services (https://marine.copernicus.eu/).

## 530 Author contributions

MM and AAC designed the study. AAC prepared the regional ocean configurations and performed the simulations. MRP prepared the parametric wind models forcings and helped to set-up the ADCIRC ocean configuration. AT and AAC designed the variance decomposition analysis. AAC did the analyses of the simulations. MM and AT supervised the project. ACC wrote



the first draft of the manuscript, and MRP wrote the parametric wind models sections. All authors contributed to manuscript
revisions and read and approved the submitted version.

**Competing interests**

The contact author declared that all the authors have a competing interest with Editor Mauricio as they all work in the same institute.

**Acknowledgments**

The authors are grateful to Clare O'Neill and Jeff Polton for sharing the NEMO barotropic code and to Chris Wilson for sharing the NEMO baroclinic Caribbean configuration. The tide gauge data used for validation are available on the GESLA website (at www.gesla.org). Analyses were carried out with Python (utide and statsmodels packages). The unstructured mesh has been designed with SMS software.

**Financial support**

AAC and AT would like to thank the Government of Cantabria through the FENIX Project GFLOOD. MM and MRP acknowledge the financial support from the ThinkInAzul programme, with funding from 761 European Union NextGenerationEU/PRTR-C17.I1 and the Comunidad de Cantabria. AT acknowledges financial support from the Ministerio de Ciencia e Innovación (MCIN/AEI and NextGenerationEU/PRTR) through the Ramon y Cajal Programme (RYC2021-030873-I).

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
