# Peer review of "Regional modelling of extreme sea levels induced by hurricanes"

_Natural Hazards and Earth System Sciences, 2024_

## Author Comment (AC1)

05.09.2024

Answer to Coleman Blakely:

The authors thank the reviewer for his assessment and constructive comments. We believe that addressing the raised issues have improved the manuscript. The point-by-point answers are provided in blue in the following.

**Main comments:**

This manuscript details an examination of two different hydrodynamic models (ADCIRC and NEMO) and their applications in modeling of storm surge due to tropical cyclones. Tropical cyclones were modelled both using ERA5 wind reanalyses and a selection of parametric wind models. Sensitives between ADCIRC and NEMO, as well as sensitivities to wind model and drag law, are evaluated. The results of the various experiments indicate that storm surge, defined as the non-tidal residual in this study, are relatively insensitive to the use of ADCIRC vs NEMO. However, there are some differences when comparing wind product used. This study shows that the reanalysis winds generally outperform the parametric wind models. Overall, I believe that this manuscript presents some useful information with regards to modeling storm surge caused by tropical cyclones. The experiments were clearly carefully designed to examine the sensitivities of interest. The combination of an inter-model as well as an intra-model comparison was particularly interesting. I do have some comments regarding scientific questions and experiments as detailed in the "Specific Comments" section; however, I think the scientific merit of this manuscript is generally sound. While the manuscript as a whole is generally interesting and explores some useful/impactful questions, there are numerous grammatical/typographical issues that make the story told hard to follow. I have detailed some specific examples of readability issues in the "Technical Corrections" section; however, these examples should not be taken to be a comprehensive list of typographical problems. I strongly suggest a thorough editing of the manuscript to improve readability and accessibility. I recommend major revisions, primarily related to presentation quality, and look forward to reading a revised version of this manuscript.

Thank you for your comment. The entire manuscript has been revised by the authors and by a professional editing service in scientific English. We believe that the new version provided has improved this issue.

**Specific comments**

- Section 2: When describing the selection of tide gauges used in the evaluation of storms you note that you select gauges that are within a 300 km radius of the hurricane in lines 91-92. In lines 95-96 you note that gauges registering storm surges of less than 15 cm are also excluded. Why not simply use the surge-based criteria alone rather than also having the distance criteria? I just find the distance criteria to be rather arbitrary whereas the surge criteria (assuming you base the selection off of observations instead of model results) is more rational.

Thank you for your comment. The distance criterion is used in this study to ensure that the selected storm surges are induced by hurricanes, rather than other processes. Additionally, it enables the validation of a surge attributed to a specific hurricane, rather than one caused by another hurricane occurring simultaneously in a different region but affecting the same tide gauge. This approach is commonly used to validate storm surges induced by tropical cyclones, as demonstrated by Dullaart et al. (2021) and Muis et al. (2019) with criteria of 250 km and 500 km, respectively. We conducted a test by removing the distance criterion, and for hurricanes Maria, Matthew, and Wilma, the selected tide gauges remained the same, with no impact on the results. However, for hurricane Irma, that had a very long life as it crossed the entire domain, we ended up selecting more tide gauges with surges over 30 cm that were not associated with Irma, and therefore missed by the parametric models, as each surge at each station is attributed to a single hurricane. Therefore, we have decided to keep the distance criterion, even though, as the reviewer stated, the magnitude criterion is prevailing over the distance. Additionally, we intend to apply the same validation method in a hindcast as part of future work. In that case, we anticipate that the distance criteria will play a more significant role.

- When describing the parametric wind models in lines 108-126, you do not touch on the various parameters within each model and what you set them to. For example, in the Chavas et. al. (2015) model there are several parameters that need to be prescribed by the user. Among these are the radiative-subsidence rate (W_cool), surface drag coefficient (C_d), and the ratio of enthalpy and momentum exchange coefficients (C_k/C_d). The other parametric wind models also have user-prescribed coefficients. You do not touch on what values you selected for these parameters. I think it would be useful to add a subsection to Section

2 where you give a short description of each parametric wind model and the values you selected in order to make the experiments herein more reproducible by others.

Thank you for your remark. Following the reviewer's suggestion, we have added subsections to Section 2 (2.1. Tide gauge data, 2.2. Atmospheric pressure and wind fields). The wind fields used in this study were generated using the default settings of each parametric wind model. The technical details and user-defined parameter values for the different models have been included in the respective description within the new subsection.

Here is the revised version of the section on parametric wind models: "Storm surges induced by hurricanes are also simulated using parametric wind models. The observations used as inputs for these models are taken from the International Best Track Archive for Climate Stewardship (IBTrACS) database (Knapp et al., 2010, 2018). The database provides at least six-hour information on the cyclone position and intensity from 1851 to the present, as well as additional variables such as the radius of maximum wind, environmental pressure, and various wind radii for recent decades. This study evaluates four recognized parametric wind models: the dynamic Holland model (DHM), the parameterization proposed by Willoughby et al. (2006), the physics-based model proposed by Chavas et al. (2015) and the generalized asymmetrical Holland model (GAHM). The DHM is an extension of the commonly used Holland model (Holland, 1980), with modifications applied by Fleming et al. (2008), to better capture dynamic processes within and around tropical cyclones. The Willoughby et al. (2006) model, also derived from the Holland model, is based on a piecewise continuous wind profile. This model is composed of analytical segments based on a power law inside the eye and two exponential decay functions outside. These segments are patched smoothly together across the radius of maximum wind using a radially varying polynomial ramp function. The required model parameters (i.e., $R1$, $R2$, $X1$, $X2$, n and $A$) are statistically estimated by these authors based on aircraft observations for nearly 500 tropical cyclones. Done et al., 2020 provide examples of successful applications of the model to be used as the forcing of marine dynamics. The Chavas et al. (2015) model is a physics-based model that mathematically merges the Emanuel (2004) and Emanuel and Rotunno (2011) solutions to integrate the outer- and inner-core wind structures of tropical cyclones, respectively. In the Chavas et al. (2015) model, several parameters need to be prescribed: the drag coefficient in the outer region (set to 0.0015), the radiative-subsidence rate (set to 2 mm/s), and the ratio of surface exchange coefficients for enthalpy and momentum (set to 1). No eye adjustment was applied to reduce the wind speed within the inner core. Wang et al., 2022 demonstrated the successful application of this model to simulate extreme sea levels. These three models assume a perfectly azimuthal symmetric structure of the wind fields (Fig. 1d, e, f), which may lead to errors in storm surge forecasting (Xie et al., 2011). The GAHM is a more recent model that also derives from the commonly used Holland model (Holland, 1980) but incorporates asymmetries in the wind field (Fig. 1c) by considering information from all available isotachs (lines of constant wind speed) in the quadrants (Gao et al., 2017; Dietrich et al., 2018; Bilskie et al., 2022). All the models provide wind velocities averaged over 1 minute at the top of the boundary layer. These velocities are first reduced by a factor of 0.9 and then adjusted by multiplication by 0.8928 to convert from 1-minute to 10-minute averages to obtain surface wind speeds. No adjustment for wind speed reduction over land has been applied. The surface pressure field is estimated based on the rectangular hyperbola approximation proposed by Schloemer (1954), including the original/generalized Holland scaling parameter $b$ as an exponent (Holland, 1980; Gao et al., 2017). The pressure drop is calculated by assuming a background pressure of 1013 mbar."

- Section 3: When describing the numerical grids of ADCIRC vs NEMO in Section 3.1, you do not note the total number of degrees of freedom of the two grids. While it is not front and center in the manuscript, you do provide wall-clock times in the appendix. Without knowing the number of mesh vertices it is hard to get a sense of the true computational cost/how it will scale with more computational power.

The number of elements for each grid has been added to the Appendix A (Tab. A1). The sentence L192 has also been modified: "A similar resolution has been chosen for ADCIRC, spanning from 3 km near the coast or in shallow water areas to 70 km in the deeper open ocean (Fig. 2b), resulting in three times fewer elements than for NEMO (Tab. A1)."

- Section 4: In lines 276-277 you note that, "ADCIRC simulations tends to underestimate the maximum [storm surge] compared to tide gauge data." Could you compute the mean error in addition to the mean absolute error to give a sense of the differences between NEMO and ADCIRC in this regard? It is hard to quantify based off of Figures 4 and 5 alone without hard numbers. Using the mean error would allow for the performance to be quantifiably evaluated instead of qualitatively. In the discussion at line 437, you also note that, "[i]n general, both models underestimate the storm surge amplitudes."

The mean bias of the maximum surges has been added for each hurricane in Figure 4 (ADCIRC vs tide gauges) and 5 (ADCIRC vs NEMO vs tide gauges). The sentence has been revised L276 to better quantify the underestimation: "Both observed and modelled highest storm surges exceed one meter for each hurricane. However, ADCIRC simulations tend to underestimate maximum storm surges compared to tide gauge data, particularly along the eastern coast of Florida (USA) and in the Caribbean Islands, resulting in a mean underestimation of at least 20% (Fig. 4)."

- You note in lines 351-355 that different parametric wind models perform are best for different storms. I feel that this might relate to the selection of parametric wind model parameters. While I do not think you need to do a sensitivity analysis of the wind model parameters (although you could), I think it is worth noting that this may be part of the issue.

Thanks for the comment. Conducting a sensitivity analysis of the wind model parameters for storm surge modelling would be very interesting. However, the large number of simulations already completed in this study made additional tests and consequent sensitivity analyses difficult. In any case, we have added the following statement L356: "A specific selection of the parameters used in the wind models for each hurricane could also enhance the accuracy of storm surge simulations (Chavas et al., 2015)."

- In lines 357-359 you say that the parametric wind models underestimate more than ERA5, could this be due to the far-afield winds not being captured in the parametric models? Does this underestimation go away if you only look at stations, for example, within the 30 knot wind radius of the center of the storm?

Thanks for the comment. To address the question, we present Figure RC1, which shows the bias in maximum storm surge relative to the distance between the tropical cyclone's eye and the tide gauges for the simulated experiments. For distances greater than 50 km, the parametric wind models consistently exhibit more substantial underestimations compared to storm surge outcomes based on ERA5 reanalysis wind fields. We attribute this behavior to the far-afield winds not being captured in the parametric models. This explanation has been added L359 to the manuscript: "While ERA5 systematically underestimates less than 50% of the time, parametric wind models tend to exhibit more substantial underestimations and occasionally miss some extreme events. This result is particularly clear when the distance between the hurricane and the tide gauge is greater than 50 km because far-afield winds are not captured in the parametric models (not shown)." On the other hand, we have validated the storm surge outcomes at specific radii, particularly for stations within twice the maximum radius (Fig. RC2), which represents 14 of the 32 stations used in the study. The results are consistent with those in Figure RC1, indicating that while underestimates are decreasing for both parametric models, overestimates are observed, resulting in overall poorer statistics compared to ERA5.

[Figure]

**Figure RC1: Bias on the maximum surge as a function of the distance to the simulated hurricane. Different colors represent the various experiments, while the markers indicate the specific hurricanes simulated: circles for Irma, squares for Wilma, upward triangles for Matthew, and downward triangles for Maria.**

[Figure]

**Figure RC2: Scatter plot of the modelled (ADCIRC) vs. observed maximum storm surges for stations within twice the maximum radius of the hurricanes: DHM (left panel), GAHM (middle panel) and ERA5 (right panel).**

- Section 6: I think that adding a description of future work would be quite valuable. For example, NEMO lacking wetting/drying capability will drastically limit its usefulness as a storm surge prediction model due to lack of ability to capture inundation of floodplains. I assume that adding a wetting/drying algorithm is part of future work but it might be useful to state here. Additionally, I think it might be useful to compare the computational performance of NEMO vs ADCIRC, especially since the manuscript highlights the usefulness of NEMO in simulating storm surges.

Thanks for the suggestion. A description of future work has been added at the end of the second paragraph of the conclusion: "These methodological insights will guide future research, especially in refining regional hindcast and projections for the tropical Atlantic. By integrating these results and addressing uncertainties in the model and the configuration development, we aim to enhance the accuracy and reliability of storm surge estimates. Furthermore, these findings may be also used in coastal impact assessment to better understand and predict hurricane-induced coastal flooding and erosion. ADCIRC provides a wetting/drying option, which is crucial for coastal impact studies, such as floodplain inundation. NEMO also has this capability (O'Dea et al., 2020) and incorporating it for future coastal studies using NEMO would be highly beneficial."

A sentence on the computational cost has been added at the end of the Discussion (Sect. 5, L424): "After testing both models, we found that NEMO and ADCIRC have comparable computation times (Tab. A1) when using similar numerical domains and resolutions. However, ADCIRC demonstrates better computational performance than NEMO because of its use of an unstructured mesh, as indicated in Table A2."

**Technical Corrections**

Lines 22-34: This paragraph is hard to follow. It seems to try and highlight two different things: 1) how tropical cyclones cause storm surge, and 2) the four storms evaluated in this manuscript. While I was able to understand the message of the paragraph, I had to read the whole thing two or three times to truly parse what was being communicated. As an example, rather than saying, "Wilma is the most intense Atlantic hurricane by lowest pressure on record, formed on October 15, 2005, reaching sustained winds of 295 km/h before making landfall in southwestern Florida on October 24, 2005.", it would be more clear to write something along the lines of, "Hurricane Wilma, which formed on Ocober 15, 2005 and made landfall in southwestern florida on October 24, is the most intense Atlantic hurricane on record as measured by atmospheric pressure." I think if might be useful to separate the description of each storm at the very least into multiple sentences. Another approach would be to remove the explicit storm descriptions and instead add more of the detail to Table 1.

The explicit storm descriptions have been removed, and additional information such as maximum sustained winds and minimum atmospheric pressure has been included in Table 1.

Lines 54-56: In the sentence starting with, "Parametric wind models," the clause, "enable to compute a large number of simulations," is grammatically incorrect.

The sentence has been modified to: "Parametric wind models represent the wind field distributions of tropical cyclones using a limited number of observations or statistical methods. The simplicity and computational efficiency of these models make them powerful tools for performing many simulations, thereby enhancing the robustness of storm surge evaluations."

Line 64: Another study that looks at the inclusion of baroclinicity in evaluating storm surge is Pringle et. al., 2019 (doi:https://doi.org/10.1029/2018JC014682).

Thanks for the reference. It has been added together with a paragraph in Sect. 4.: "Pringle et al. (2019) investigated the baroclinic contribution to storm surges using a 2DDI (depth-integrated) configuration that incorporated baroclinic effects on the free surface and depth-integrated currents without simulating them directly. They reported that for Puerto Rico (San Juan station) during Hurricanes Maria and Irma, the predicted maximum surge increased by approximately 20 cm because of baroclinic effects, which is greater than the increase observed in our study."

Lines 100-102: You describe ERA5 as having a resolution of 0.25 degrees and ERA-Interim as having a resolution of 79 km. While it is relatively simple to do the conversion I would suggest including a ballpark resolution in kilometers for ERA5, e.g., "The atmospheric variables have a horizontal resolution of 0.25 degrees (~25 km)..."

The sentence L99-100 has been modified: "The ERA5 has a horizontal resolution of 0.25 degrees (~31 km)."

Lines 108-126: A good chunk of this paragraph could be moved to the introduction, particularly the description of the history of parametric wind models. The description of IBTrACS should remain in this section. Alternatively, as mentioned above, having a separate subsection with each parametric model described in more detail would be useful.

A portion of this paragraph has been moved to the Introduction as suggested by the reviewer. The sub-sections for each parametric wind model have not been created to maintain a balance in the number and size of sections, but we have added subsections to Section 2 (2.1. Tide-gauge data, 2.2. Atmospheric pressure and wind fields). The entire section has been revised as stated in a previous comment.

Line 180: "Tidal amplitudes are relatively moderated in the region..." This is a little awkward. Perhaps clarify what you mean by "moderated".

The sentence has been changed to: "The region is microtidal with the largest tidal amplitudes reaching approximately 2 meters in the northern Surinam and Guyana, as well as in the northeastern Florida (USA)."

Line 213: "This experience is conducted with the ADCIRC..." Should be "experiment".

Corrected.

Line 235: You say that variables are described, "in the NEMO description part." Give a section number instead to make it easier to locate where this information can be found.

The part of the sentence has been updated: "[…] by the ERA5 atmospheric reanalysis (Sect. 2.2) at the air-sea interface (see variables in Tab. 3)."

These are just some examples of places where I found the readability of this manuscript to be lacking. I think that some thorough editing will really bring out the message and findings of the article and make it much more accessible to the reader.

Thank you very much for the suggestion. We have applied a professional editing service in scientific English to the entire text of the manuscript.

**References**

Bilskie, M. V., Angel, D. D., Yoskowitz, D., and Hagen, S. C.: Future Flood Risk Exacerbated by the Dynamic Impacts of Sea Level Rise Along the Northern Gulf of Mexico, Earth's Future, 10, e2021EF002414, https://doi.org/10.1029/2021EF002414, 2022.

Chavas, D. R., Lin, N., and Emanuel, K.: A Model for the Complete Radial Structure of the Tropical Cyclone Wind Field. Part I: Comparison with Observed Structure, Journal of the Atmospheric Sciences, 72, 3647–3662, https://doi.org/10.1175/JAS-D-15-0014.1, 2015.

Dietrich, J. C., Muhammad, A., Curcic, M., Fathi, A., Dawson, C. N., Chen, S. S., and Luettich, R. A.: Sensitivity of Storm Surge Predictions to Atmospheric Forcing during Hurricane Isaac, J. Waterway, Port, Coastal, Ocean Eng., 144, 04017035, https://doi.org/10.1061/(ASCE)WW.1943-5460.0000419, 2018.

Done, J. M., Ge, M., Holland, G. J., Dima-West, I., Phibbs, S., Saville, G. R., and Wang, Y.: Modelling global tropical cyclone wind footprints, Natural Hazards and Earth System Sciences, 20, 567–580, https://doi.org/10.5194/nhess-20-567-2020, 2020.

Dullaart, J. C. M., Muis, S., Bloemendaal, N., Chertova, M. V., Couasnon, A., and Aerts, J. C. J. H.: Accounting for tropical cyclones more than doubles the global population exposed to low-probability coastal flooding, Commun Earth Environ, 2, 135, https://doi.org/10.1038/s43247-021-00204-9, 2021.

Emanuel, K.: Tropical cyclone energetics and structure, in: Atmospheric Turbulence and Mesoscale Meteorology, edited by: Fedorovich, E., Rotunno, R., and Stevens, B., Cambridge University Press, 165–192, https://doi.org/10.1017/CBO9780511735035.010, 2004.

Emanuel, K. and Rotunno, R.: Self-Stratification of Tropical Cyclone Outflow. Part I: Implications for Storm Structure, Journal of the Atmospheric Sciences, 68, 2236–2249, https://doi.org/10.1175/JAS-D-10-05024.1, 2011.

Fleming, J. G., Fulcher, C. W., Luettich, R. A., Estrade, B. D., Allen, G. D., and Winer, H. S.: A real time storm surge forecasting system using ADCIRC: 10th International Conference on Estuarine and Coastal Modeling, Estuarine and Coastal Modeling - Proceedings of the 10th International Conference, 893–912, https://doi.org/10.1061/40990(324)48, 2008.

Holland, G. J.: An Analytic Model of the Wind and Pressure Profiles in Hurricanes, Monthly Weather Review, 108, 1212–1218, https://doi.org/10.1175/1520-0493(1980)108<1212:AAMOTW>2.0.CO;2, 1980.

Knapp, K. R., Kruk, M. C., Levinson, D. H., Diamond, H. J., and Neumann, C. J.: The International Best Track Archive for Climate Stewardship (IBTrACS): Unifying Tropical Cyclone Data, Bulletin of the American Meteorological Society, 91, 363–376, https://doi.org/10.1175/2009BAMS2755.1, 2010.

Muis, S., Lin, N., Verlaan, M., Winsemius, H. C., Ward, P. J., and Aerts, J. C. J. H.: Spatiotemporal patterns of extreme sea levels along the western North-Atlantic coasts, Sci Rep, 9, 3391, https://doi.org/10.1038/s41598-019-40157-w, 2019.

O'Dea, E., Bell, M. J., Coward, A., and Holt, J.: Implementation and assessment of a flux limiter based wetting and drying scheme in NEMO, Ocean Modelling, 155, 101708, https://doi.org/10.1016/j.ocemod.2020.101708, 2020.

Pringle, W. J., Gonzalez-Lopez, J., Joyce, B. R., Westerink, J. J., and van der Westhuysen, A. J.: Baroclinic Coupling Improves Depth-Integrated Modeling of Coastal Sea Level Variations Around Puerto Rico and the U.S. Virgin Islands, Journal of Geophysical Research: Oceans, 124, 2196–2217, https://doi.org/10.1029/2018JC014682, 2019.

Wang, S., Lin, N., and Gori, A.: Investigation of Tropical Cyclone Wind Models With Application to Storm Tide Simulations, Journal of Geophysical Research: Atmospheres, 127, e2021JD036359, https://doi.org/10.1029/2021JD036359, 2022.

Willoughby, H. E., Darling, R. W. R., and Rahn, M. E.: Parametric Representation of the Primary Hurricane Vortex. Part II: A New Family of Sectionally Continuous Profiles, Monthly Weather Review, 134, 1102–1120, https://doi.org/10.1175/MWR3106.1, 2006.

Xie, L., Liu, H., Liu, B., and Bao, S.: A numerical study of the effect of hurricane wind asymmetry on storm surge and inundation, Ocean Modelling, 36, 71–79, https://doi.org/10.1016/j.ocemod.2010.10.001, 2011.

---

## Author Comment (AC2)

05.09.2024

Answer to Referee 2:

The authors thank the reviewer for his/her comments. The point-by-point answers to the review are provided in blue in the following.

**Main comments:**

This manuscript focuses on the performance of NEMO and ADCIRC ocean models in simulating storm surges in tropical east Atlantic. It concludes that both ocean models can simulate storm surge in a similar way. It also concludes that ERA5 atmospheric reanalysis forcing gives better results than parametric wind models and also that inclusion of baroclinic processes in the simulation improves the result significantly. Different wind stress parametrizations, and the surge interaction with tide and mean sea level have shown little to minimal impact. Globally I would say that this study is well conducted, and gives some interesting results. However, I would suggest a general review to improve the flow and readability. Sometimes it gets hard to follow.

Thank you for the comment. The entire manuscript has been revised by the authors and by a professional editing service in scientific English. We believe that the new version provided has improved this issue.

**Specific comments**

- In lines 219-230 you describe how you investigated the impact of wind stress parameterizations. I find this section very hard to follow. I also would say that is poor in terms of content. I would re-write this part describing both equations in more detail, especially Eq.1, that has no description of the variables whatsoever.
- I appreciate the fact that you performed a different simulation with a variable Charnock parameter depending on wave parameters. You should explain why you performed this additional simulation, mentioning why sea roughness is dependent on wave parameters, explaining what variables have an impact, and cite the authors that investigated this process.

Thank you for your comments. We have modified the explanation of the wind stress parameterizations tested as follows (L221-230):

"The barotropic configuration of NEMO is also used to investigate the impact of wind stress parameterization on storm surges, taking advantage of the flexibility of NEMO in modifying the code. This study compares the S&B (Smith and Banke, 1975) scheme (Eq. (1)) with the Charnock formulation (Charnock, 1955) (Eq. (2)). In the S&B scheme, the wind stress $\tau$ is calculated using a simple formulation for the drag coefficient $C_D$, which represents the drag force exerted by the wind on the water surface, as follows:

$$(1) \quad \tau = \rho_a C_D U^2 \text{ with } C_D = (0.75 + 0.067|U|)e^{-3}$$

where $\rho_a$ is the air density and U is the 10 m wind speed.

The Charnock relationship is a semiempirical formula that involves a more complex calculation accounting for changes in surface roughness with wind speed as follows:

$$(2) \quad \tau = \rho_a u_*^2 \text{ with } z_0 = \frac{\alpha u_*^2}{g}$$

where $z_0$ is the roughness length, α is the dimensionless Charnock parameter, $u_*$ is the friction velocity and g is gravity. In general, the Charnock parameter α is generally assumed to be constant in the formulation of sea surface roughness (Eq. (2)). For example, in the standard NEMO code, it is kept constant in space and time, equal to 0.018. In reality, this parameter varies with sea surface roughness and is influenced by various wave parameters, such as wave age, wave steepness and the presence of sea foam, especially under high wind conditions, as suggested by numerous studies published in recent decades (Janssen, 1989; Moon et al., 2004; Pineau-Guillou et al., 2020; Wu et al., 2024). An additional simulation has therefore been performed using a variable Charnock parameter derived from ERA5 reanalysis outputs, which depend on wave conditions (Riverside Technology, 2015)."

- Also, I could not find (Smith and Banke, 1975) on your reference list. There could be others missing. I recommend to check the list very carefully.

Thank you. The reference Smith and Banke (1975) has been added and the whole reference list has been checked.

**References**

Charnock, H.: Wind stress on a water surface, Quarterly Journal of the Royal Meteorological Society, 81, 639–640, https://doi.org/10.1002/qj.49708135027, 1955.

Janssen, P. A. E. M.: Wave-Induced Stress and the Drag of Air Flow over Sea Waves, 1989.

Moon, I.-J., Ginis, I., and Hara, T.: Effect of surface waves on Charnock coefficient under tropical cyclones, Geophysical Research Letters, 31, https://doi.org/10.1029/2004GL020988, 2004.

Pineau-Guillou, L., Bouin, M.-N., Ardhuin, F., Lyard, F., Bidlot, J.-R., and Chapron, B.: Impact of wave-dependent stress on storm surge simulations in the North Sea: Ocean model evaluation against in situ and satellite observations, Ocean Modelling, 154, 101694, https://doi.org/10.1016/j.ocemod.2020.101694, 2020.

Riverside Technology, I., and Aecom: Mesh Development, Tidal Validation, and Hindcast Skill Assessment of an ADCIRC Model for the Hurricane Storm Surge Operational Forecast System on the US Gulf-Atlantic Coast, https://doi.org/10.17615/4z19-y130, 2015.

Smith, S. D. and Banke, E. G.: Variation of the sea surface drag coefficient with wind speed, Quarterly Journal of the Royal Meteorological Society, 101, 665–673, https://doi.org/10.1002/qj.49710142920, 1975.

Wu, L., Sahlée, E., Nilsson, E., and Rutgersson, A.: A review of surface swell waves and their role in air–sea interactions, Ocean Modelling, 190, 102397, https://doi.org/10.1016/j.ocemod.2024.102397, 2024.